# CameraNoise: Learning Precise Camera Control with Video Diffusion in Noise Space

## Abstract

Controlling camera pose in video diffusion models is essential for generating realistic videos, yet existing approaches struggle to achieve precise control. Methods that directly inject numerical camera parameters into the diffusion backbone often fail to capture subtle viewpoint variations and lead to structural distortions or visual artifacts. To overcome these limitations, we propose **CameraNoise**, a temporally coherent stochastic representation warped from camera intrinsic and extrinsic parameters. Unlike conventional approaches, CameraNoise embeds camera poses directly into the noise space. This makes our approach independent of scene appearance while faithfully encoding camera motion. Specifically, we introduce a novel Geometry-guided Reprojection Flow along with a CameraNoise warping algorithm, which jointly preserves the Gaussian prior of diffusion and ensures consistent noise propagation under camera transformations. By integrating CameraNoise into the diffusion process, our framework delivers stable and high-quality videos with precise camera control. Extensive experiments on the RealEstate10K benchmark demonstrate that our approach significantly outperforms prior methods in both fidelity and controllability. Anonymous project page is at https://lizaigc.github.io/.

## 1 Introduction

Camera-controllable video generation has emerged as a core research direction, driven by recent advances in general video diffusion models. Unlike conventional video synthesis, this task aims to generate videos that follow a specific camera trajectory through precise control of camera poses, *i.e.,* the intrinsic and the extrinsic matrices (He et al., 2025b; Zheng et al., 2025). Such control is crucial for applications from personalized video creation and virtual environments to filmmaking, where fine-grained manipulation of viewpoint and motion ensures realism and flexibility.

Despite its broad potential in real-world applications, camera-controllable video generation remains challenging. Achieving realistic results requires temporal smoothness across frames, consistent scene geometry under diverse camera motions, and generalization to arbitrary trajectories. To this end, prior methods typically encode camera poses as Plücker embeddings (He et al., 2025a; Bahmani et al., 2025) or linear feature vectors (Wang et al., 2024b), and inject these numerical representations into the diffusion backbone. While effective, this strategy suffers from two fundamental drawbacks: **1**) they often provide an imprecise representation of camera pose (as shown in cases (a) and (b) in Fig. 1), particularly when capturing subtle lens movements and variations in camera speed; **2**) they tend to produce structural distortions and texture discontinuities under new viewpoints.

These issues indicate that feature injection alone is insufficient for reliable camera control. Motivated by recent advances in noise warping (Daras et al., 2024; Chang et al., 2024; Burgert et al., 2025) that leverage optical flow to construct temporally correlated noise fields for motion generation, we propose embedding viewpoint information directly into the noise space as a more principled solution. This ensures that camera conditioning persists throughout the entire generative process rather than being limited to intermediate activations. However, most existing methods (Burgert et al., 2025; Chang et al., 2024), which rely on optical flow derived from object motion, inevitably encode contours and appearance details of the original scene. Consequently, motion information becomes entangled with appearance priors, which can conflict with textual conditions during inference and lead to generation failures (as shown in cases (c) and (d) in Fig. 1).

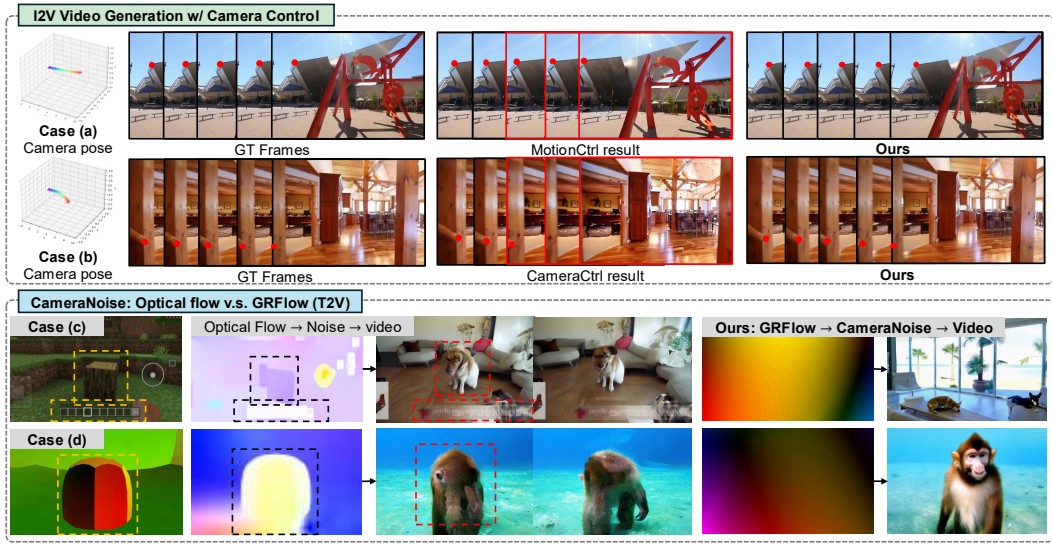

Figure 1: We highlight two key observations. (**1**): Shown in case (a) and (b), camera trajectories visualized by stacked frames with *red anchor points* show that MotionCtrl (Wang et al., 2024b) and CameraCtrl (He et al., 2025a) provide only limited precision, while our method achieves accurate control. (**2**): Comparing optical flow and our GRFlow for CameraNoise generation. Optical flow entangles appearance with motion, causing artifacts (red dashed boxes). By contrast, GRFlow enables appearance-agnostic warping, yielding visually correct and coherent video content. Camera motion templates are obtained from Minecraft (case (c)) and Unreal Engine (case (d)). The textual prompts for cases (c) and (d) are "*Two dogs in a room*" and "*A monkey in the water*", respectively.

In this paper, we propose a novel camera control framework that introduces a warped CameraNoise to enable precise control and high-quality video generation directly in the noise space. Inspired by prior noise warping methods (Chang et al., 2024; Burgert et al., 2025), our CameraNoise retains the advantages of Gaussian noise representations for capturing temporal relations of camera poses, but crucially, it is constructed to be independent of appearance information. This makes CameraNoise inherently appearance-agnostic, allowing it to be fully decoupled from object textures and motion. By injecting CameraNoise into the diffusion process, we achieve camera-controllable video generation while avoiding the semantic conflicts commonly observed in optical-flow-based approaches.

To construct CameraNoise, we introduce Geometry-guided Reprojection Flow (GRFlow), a flow representation that disentangles camera motion from visual content. Unlike optical flow, GRFlow relies solely on camera parameters to characterize pixel displacements across frames. Concretely, we project per-frame camera parameters onto a grid plane and employ Lie algebra optimization to reduce jitter caused by camera pose estimation errors. Building upon GRFlow, we formulate noise warping as a partial differential equation problem and solve it via a bipartite graph. This formulation establishes a one-to-one mapping among camera poses, GRFlow, and CameraNoise. The resulting CameraNoise is then combined with standard Gaussian noise and injected into the diffusion process for camera control. To further enhance inference robustness, we introduce dynamic perturbations to camera extrinsics during training. We evaluate our method on the RealEstate10K dataset (Zhou et al., 2018) across both image- and text-to-video tasks. Experimental results show that our approach achieves more accurate camera control, higher video quality, and stronger visual performance than prior methods, effectively addressing the challenge of imprecise camera control.

In short, our work makes the following four contribution:

- We propose a novel camera-controllable diffusion framework that learns camera motion directly from temporal noise. We term this representation CameraNoise, a warped Gaussian signal that preserves temporal correlations derived from camera parameters.

- We propose an appearance-agnostic Geometry-guided Reprojection Flow (GRFlow) and a corresponding reprojection algorithm for constructing CameraNoise, which disentangles camera motion from scene appearance and prevents semantic conflicts during synthesis.

- We propose formulating CameraNoise warping from GRFlow as a partial differential equation problem, achieving a one-to-one mapping from camera poses to CameraNoise, which ensures precise controllability of the camera motion.

- We conduct extensive evaluation to demonstrate the effectiveness of our method, highlighting its ability to generate high-quality videos with precise camera control.

## 2 RELATED WORK

**Video diffusion.** Since the success and the popularity of the Diffusion Model (Ho et al., 2020), many works pay a lot of effort into achieving video generation (Gu et al., 2023; Chen et al., 2024; Wang et al., 2023b; Zhao et al., 2025; Ni et al., 2024) with deep learning models. In the early stage, video diffusion relies on the pre-trained Stable Diffusion model for image synthesis (Guo et al., 2023; Wang et al., 2023a; Chen et al., 2023). They direct the temporal layer to the SD model to learn the temporal information. Due to the limited performance of temporal consistency, this structure is replaced by models (Gu et al., 2023; Blattmann et al., 2023; Zhao et al., 2025) with 3D convolution and attention layers. Recently, with the growing popularity of DiT model (Peebles & Xie, 2023), several DiT-based diffusion approaches (Yang et al., 2024; Xu et al., 2024; Kong et al., 2024; Wan et al., 2025; Gao et al., 2025) have become mainstream in video generation.

**Camera controllable video generation.** Camera-controllable diffusion model aims to ensure that generated videos follow the motion trajectories defined by camera parameters. This property is inherent to nearly all videos, including those captured or generated under static camera conditions. To achieve such control, recent methods (He et al., 2025a; Wang et al., 2024b) typically encode numerical camera features from intrinsic and extrinsic matrices and directly inject them into diffusion blocks via ControlNet or adapter structures. In addition to using camera parameter matrices, methods such as Gen3C (Ren et al., 2025) and AC3D (Bahmani et al., 2025) leverage 3D information to improve scene consistency and coherence. A distinct challenge addressed by models like ReCapture (Zhang et al., 2025) and ReCamMaster (Bai et al., 2025) is the re-rendering of existing videos from new camera perspectives. However, existing methods, whether based on camera parameters or 3D conditions, inject numerical features into diffusion models to learn the relationship between parameters and video. This often results in limited sensitivity to camera control and difficulty explicitly distinguishing between fast and slow motion. Recent studies (Chang et al., 2024; Burgert et al., 2025) have demonstrated that incorporating temporal information into the noise space can guide video generation toward specific motion patterns. Inspired by this, we propose CameraNoise, a novel approach that directly controls camera trajectories in the noise space.

**Camera pose estimation.** When generating videos from camera parameters, control conditions typically rely on camera estimation algorithms (Melekhov et al., 2017; Camposeco et al., 2018; Schonberger & Frahm, 2016) because most online videos including camera control templates lack explicit camera parameter annotations. Even the RealEstate10K dataset (Zhou et al., 2018) uses model-based estimation to generate its annotations. COLMAP (Schonberger & Frahm, 2016) is the most widely used traditional pipeline, comprising multiple stages such as image matching, triangulation, and bundle adjustment. Among recent methods, VGGT (Wang et al., 2025) achieves state-of-the-art results by directly estimating camera parameters through extensive parameter fitting. Given these advantages, we adopt VGGT for camera parameter estimation and model evaluation.

## 3 METHOD

Our primary objective is to enable camera-controllable video generation while mitigating imprecise control from numerical embedding injections and distortions during generation. As shown in Fig. 2, our framework encodes camera motion patterns directly in the noise space via GRFlow and CameraNoise, enabling precise viewpoint control without interfering with the denoising process.

### 3.1 GEOMETRY-GUIDED REPROJECTION FLOW

Since CameraNoise models pixel motion across frames, we first introduce an appearance-agnostic Geometry-guided Reprojection Flow (GRFlow), denoted as $\mathcal{G}_r$, along with its corresponding reprojection algorithm. GRFlow defines a precise mapping from camera poses to geometric trans-

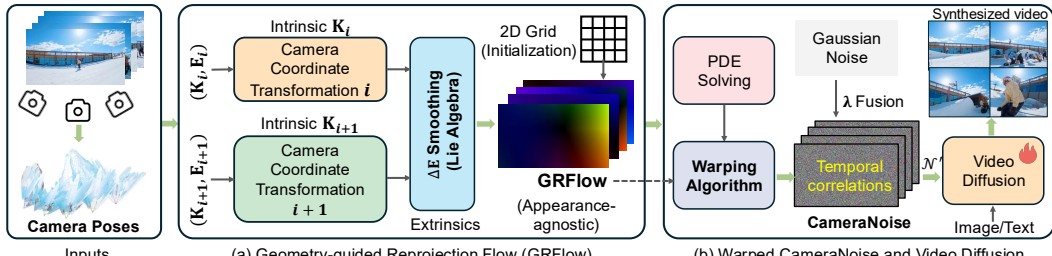

Figure 2: Overview of our framework. We introduce CameraNoise, a controlled noise signal that encodes temporal correlations of camera poses into video diffusion. Our method is constructed via Geometry-guided Reprojection Flow (GRFlow) and a Gaussian-preserving warping algorithm, and injected into the video diffusion to enable precise viewpoint control. We use bold green arrows to illustrate the flow of control signals from camera poses to the synthesized video.

formations, effectively decoupling motion information from appearance cues, a problem that typically arises in optical-flow-based methods. Specifically, we take camera parameters $[\mathbf{K}_i, \mathbf{E}_i]$ for the source frame and $[\mathbf{K}_{i+1}, \mathbf{E}_{i+1}]$ for the target frame, where $\mathbf{K} \in \mathbb{R}^{3\times3}$ denotes the intrinsic matrix and $\mathbf{E} = [\mathbf{R}; \mathbf{t}] \in \mathbb{R}^{3\times4}$ the extrinsic matrix. Together, the pair $(\mathbf{R}, \mathbf{t})$ fully determines the camera's imaging process in the world coordinate system $\mathbf{R}^w$.

**Reprojection using camera poses.** Let the 2D image plane be a discrete sampling of a continuous manifold $\mathcal{I} \subset \mathbb{R}^2$, represented by the grid $\Omega \subset \mathcal{I}$. The camera extrinsic pose $\mathbf{E} = [\mathbf{R}; \mathbf{t}]$ is represented as an element of $\mathrm{SE}(3)$, the Special Euclidean group, which acts on the 3D scene manifold $\mathcal{S} \subset \mathbb{R}^3$. We define the GRFlow as:

$$\mathcal{G}_r : \Omega \times \mathrm{SE}(3) \to \Omega, (x, y; \mathbf{E}_i, \mathbf{E}_{i+1}) \mapsto (x', y'), \tag{1}$$

that encodes the effect of rigid camera motion on 2D pixel positions without relying on scene appearance (Proposition 1). We denote $(x, y)$ as a pixel from the source frame, which corresponds to the pixel $(x', y')$ on the target frame. Each pixel at $(x, y)$ is represented in homogeneous form as $\Omega_{x,y} = [x, y, 1]^\top \in \mathbb{P}^2$, where $\mathbb{P}^2$ means the 2D projective space. These pixels define rays in the camera coordinate via the inverse intrinsic matrix. To reconstruct the 3D points, we rely solely on camera poses and back-project to the camera coordinate frame using a constant pseudo-depth $d$:

$$\mathbf{p}_{x,y} = \tilde{\Omega}_{x,y} \cdot d; \ \tilde{\Omega}_{x,y} = \Omega_{x,y} \cdot \mathbf{K}_i^{-1} \in \mathbb{R}^3. \tag{2}$$

We formalize the Eq. (2) as a lifting map that lifts the 2D point into 3D along its viewing ray:

$$\ell : \Omega \to \mathbb{R}^3, \quad \ell(x, y) = d \cdot \mathbf{K}_i^{-1}[x, y, 1]^\top. \tag{3}$$

Then, we extend the 3D point to homogeneous coordinates as: $\tilde{\mathbf{p}}_{x,y} = [\mathbf{p}_{x,y}, 1]^\top \in \mathbb{P}^3$ and map it to the target camera coordinate using the $\Delta\mathbf{E} = \mathbf{E}_i^\top \times \mathbf{E}_{i+1} \in \mathrm{SE}(3)$, which denotes the relative transformation between source frame and target frame.

**$\Delta\mathbf{E}$ smoothing via Lie algebra.** In practice, errors in camera pose estimation are inevitable, leading to discontinuities in $\Delta\mathbf{E}$ and causing irregular jitter in GRFlow. Since the rotation matrix $\mathbf{R} \in \mathrm{SO}(3)$ is subject to nonlinear constraints, direct optimization in the matrix space makes it difficult to preserve the group structure. To address this issue, we propose a smoothing function for $\Delta\mathbf{E}$ based on Lie algebra, which maps nonlinear transformation matrices into a linear space for processing. Specifically, we extract the rotational vector $\omega_i \in \mathrm{so}(3)$ from $\mathbf{R}_i$ and compute the rotation angle:

$$\cos\theta_i = (\mathrm{tr}(\mathbf{R}_i) - 1)/2 \implies \theta_i = \arccos(\cos\theta_i). \tag{4}$$

We calculate the unit vector of the rotation axis $\mathbf{k}_i$ and get the rotational vector as: $\omega_i = \theta_i \times \mathbf{k}_i$. The extrinsic matrix $\mathbf{E}_i$ is then represented with Lie Algebra vector as $\xi_i = [\boldsymbol{\omega}_i, \boldsymbol{t}_i] = [\omega_{i,x}, \omega_{i,y}, \omega_{i,z}, t_{i,x}, t_{i,y}, t_{i,z}] \in \mathrm{se}(3)$, where $\{\omega_{i,x}, \omega_{i,y}, \omega_{i,z}\}$ are rotate vectors and $\{t_{i,x}, t_{i,y}, t_{i,z}\}$

are translations. For small rotations ($\theta \approx 0$), we set $\omega_i = 0$. This Lie algebra representation enables smoothing of the trajectory using an exponentially weighted moving average with factor $\alpha$:

$$\xi_i^{\text{smooth}} = \begin{cases} \xi_i, & i = 1 \\ \alpha \cdot \xi_i + (1 - \alpha) \cdot \xi_{i-1}^{\text{smooth}}, & i > 1. \end{cases} \tag{5}$$

To incorporate the smoothed parameters into the GRFlow, we apply the exponential map to convert the $\xi_i^{\text{smooth}}$ back into a transformation matrix $\Delta \mathbf{E}$ in the SE(3) space.

**GRFlow construction.** The transformed 3D point is computed by matrix multiplication in homogeneous coordinates: $\tilde{\mathbf{p}}'_{x,y} = \Delta \mathbf{E} \times \tilde{\mathbf{p}}_{x,y}$. This operation is equivariant under SE(3), meaning that the transformation preserves rigid-body geometry:

$$\|\mathbf{p}'_1 - \mathbf{p}'_2\|_2 = \|\mathbf{p}_1 - \mathbf{p}_2\|_2, \quad \forall \mathbf{p}_1, \mathbf{p}_2 \in \mathbb{R}^3. \tag{6}$$

Subsequently, we reproject points onto the 2D image plane of the target camera: $\Omega'_{x,y} = \mathbf{K}_{i+1} \times \tilde{\mathbf{p}}'_{x,y} \in \mathbb{P}^2$, and obtain the Cartesian coordinates $(x', y')$ by normalizing homogeneous coordinates. The particle flow on the source image, $\mathcal{F}_{x,y}$, is then defined as the displacement between the target and source pixel locations: $\mathcal{F}_{x,y} = (x', y') - (x, y)$. Iterating this flow across all frames generates the full GRFlow, which pushes forward the pixel coordinates through the rigid-body action:

$$\mathcal{G}_r = \{\mathcal{F}_{x,y}^{i \to i+1} \mid (x, y) \in \Omega, i = 1, \dots, T - 1\}. \tag{7}$$

Furthermore, we show that GRFlow serves as an approximation of continuous flow (Proposition 2), and we provide a formal proof in the Appendix A.

> **Proposition 1** (Appearance Agnostic). *GRFlow is computed only from camera poses, so it captures geometric motion while remaining independent of visual appearance.*
> **Proposition 2** (Approximation of Continuous Flow). *Under smoothed camera motion and typical latent diffusion resolution, GRFlow converges pointwise to the continuous flow map, with discretization bounding the error.*

### 3.2 THE CAMERANOISE REPRESENTATION

The main purpose of CameraNoise is to encode temporal priors of camera motion into Gaussian noise using GRFlow. Specifically, GRFlow propagates the initial noise such that it remains temporally consistent while preserving its Gaussian distribution. However, a naive interpolation between adjacent frame pixels would violate this prior (see Appendix B.1). To address the issue, we introduce a novel warping algorithm to generate CameraNoise, which formulates noise propagation as the discrete solution of advection *Partial Differential Equations (PDEs)*. Furthermore, we incorporate a density correction mechanism to uniformly accommodate diverse camera motion patterns.

Given an initial noise frame $q_t \in \mathbb{R}^{H \times W}$ and the advection vectors for each grid point in GRFlow $\mathcal{G}_r$, we aim to generate the subsequent noise frame $q_{t+1}$. The propagation from $q_t$ to $q_{t+1}$ is constrained by the following principles: **1**) Temporal continuity: the next-frame noise depends solely on the state of the previous frame; **2**) Preservation of the Gaussian prior: the noise distribution remains a standard Gaussian. We formulate noise propagation as a linear advection PDE:

$$\frac{\partial \rho(x, t)}{\partial t} + v(x, t) \cdot \nabla \rho(x, t) = 0, \quad v(x, t) = \mathcal{G}_r^t, \tag{8}$$

where $\rho(x, t)$ denotes the probability density of the noise field at position $x$ and time $t$, and $v(x, t)$ represents the velocity field given by $\mathcal{G}_r$. This equation describes the advection of the noise density $\rho$ by the velocity field $v$, where volume changes modulate the local noise concentration. To solve the PDE on discrete frames, we adopt a bipartite graph formulation combined with a density-weighted discretization strategy. In this graph, nodes on the left correspond to positions $\rho_t(y)$ in the current noise frame, while nodes on the right represent positions $\rho_{t+1}(x)$ in the subsequent frame. Edges

$(y \to x)$ encode the transport paths determined by GRFlow. On a discrete grid, the continuous flow is represented as correspondences between pixels, which can be categorized into two motion types: expansion and contraction. For the edge weights $w(x, y)$, we compute either the local flow density or the determinant of the Jacobian matrix $\det J(x)$ (details in Appendix C), which quantifies the local degree of expansion and contraction in the noise field. For the backward flow in GRFlow, we directly use $-\mathcal{G}_r$. The divergence term $\nabla \cdot (\rho v)$ in the PDE is discretized as a mass-transfer operation over the graph edges. Then, density normalization is computed as:

$$\rho_{t+1}(x) = \sum_{y \mapsto x} w(x, y)\rho_t(y) \, / \sum_{y \mapsto x} w(x, y). \tag{9}$$

By leveraging a bipartite graph and sparse matrix-based transport, we reformulate the PDE as a linear algebra problem defined on the graph. This warping algorithm circumvents the complexity of directly solving the PDE, yielding a stable and computationally efficient discrete solution. By iterating this process across frames, we generate a temporally consistent noise sequence, which we denote as CameraNoise. We further demonstrate that the mapping from camera poses to CameraNoise is one-to-one (Proposition 3), with the corresponding proof provided in Appendix B.2.

> **Proposition 3** (One-to-One Mapping). *By the PDE uniqueness theorem, each $\rho$ has a unique solution $\rho(x)$, making the mapping from GRFlow to CameraNoise bijective and ensuring a one-to-one correspondence between camera poses and CameraNoise.*

### 3.3 VIDEO DIFFUSION MODEL WITH CAMERANOISE

In this subsection, we introduce CameraNoise into the diffusion model to achieve camera-controllable video generation. After undergoing GRFlow reprojection and PDE solving, CameraNoise encodes temporally coherent information in the noise space that is equivalent to camera motion. This enables it to guide the motion direction of the generated frames during inference, thereby simulating camera movements. To integrate CameraNoise $\mathcal{N}_C$ into the diffusion process, we fuse it with random Gaussian noise $\mathcal{N}$ during training with:

$$\mathcal{N}' = (\mathcal{N} \cdot \lambda + \mathcal{N}_C \cdot (1 - \lambda)) / \sqrt{\lambda^2 + (1 - \lambda)^2}, \tag{10}$$

where $\lambda$ is a mixing coefficient that regulates the relative contribution of the two components. During training, we randomly sample $\lambda$ from the interval $[0, 1]$ for each training case. This strategy ensures that the model remains sensitive to both the standard Gaussian distribution and CameraNoise. Because CameraNoise encodes camera-specific information, directly replacing the standard Gaussian with CameraNoise in the diffusion model would compromise the model's prior knowledge. During inference, the fusion coefficient is set to zero or a small value, ensuring stable control.

Moreover, in the training phase, we introduce a Dynamic Scaling Training (DST) strategy for extrinsic matrices to enhance the robustness of camera parameter estimation. For rotate matrix $\mathbf{R}$ in the extrinsic, we denote the $\mathbf{R} = \exp(\theta[\mathbf{u}]_\times)$, with Rodrigues function (Dai, 2015), where $\mathbf{u} \in \mathbb{R}^3$, $\|\mathbf{u}\|$ denotes the rotary axis, and $\theta \in \mathbb{R}$ represents the rotation angle. The $[\mathbf{u}]_\times \in \mathbb{R}^{3\times3}$ is the antisymmetric matrix of $\mathbf{u}$. We denote the rotational vector $\mathbf{r}$ as: $\mathbf{r} = \theta\mathbf{u}$ and the rotate matrix is:

$$\mathbf{R} = \exp(\theta[\mathbf{u}]_\times) = \exp([\mathbf{r}]_\times); \text{with } \mathbf{r} = (r_x, r_y, r_z)^\top, \theta = \sqrt{r_x^2 + r_y^2 + r_z^2}. \tag{11}$$

We define the scale factor $\eta$ for $\mathbf{r}$ as $\mathbf{r}' = \epsilon \times \mathbf{r} = (\eta \times \theta) \times \mathbf{u}$. Then, we can obtain rescaled $\mathbf{R}'$:

$$\mathbf{R}' = \exp([\mathbf{r}']_\times) = \exp((\eta \cdot \theta) \cdot [\mathbf{u}]_\times). \tag{12}$$

We maintain the translation part $\mathbf{t}$ and update the extrinsic parameter as: $\mathbf{E} = [\mathbf{R}'; \mathbf{t}]$. In practice, we control the factor $\eta$ manually, and we set $\eta \in (0.9, 1.1)$ for diffusion training. This scaling strategy improves model performance during inference with estimated camera poses, particularly enhancing robustness across varying rotation angles.

Table 1: Quantitative comparisons with MotionCtrl (Wang et al., 2024b), CameraCtrl (He et al., 2025a), and AC3D (Bahmani et al., 2025) methods on RealEstate100 test set, including Text-to-Video (T2V) and Image-to-Video (I2V) generation.

| Methods | Aesthetic Quality ↑ | Imaging Quality ↑ | Motion Smoothness | Dynamic Degree | TransErr ↓ | RotErr ↓ | LPIPS ↓ | FVD ↓ |
|---|---|---|---|---|---|---|---|---|
| MotionCtrl-T2V | 0.521 | 0.731 | 0.980 | 0.10 | 0.372 | 0.476 | **0.468** | 661.04 |
| CameraCtrl-T2V | 0.403 | 0.706 | 0.974 | 0.26 | 0.344 | 0.483 | 0.556 | 909.58 |
| AC3D | 0.476 | 0.613 | **0.993** | 0.15 | 0.391 | 0.472 | 0.634 | 800.79 |
| **Ours-T2V** | **0.549** | **0.740** | **0.993** | **0.53** | **0.232** | **0.436** | 0.528 | **437.54** |
| MotionCtrl-I2V | 0.414 | 0.654 | 0.992 | 0.33 | 0.445 | 0.606 | 0.312 | 321.79 |
| CameraCtrl-I2V | 0.482 | 0.672 | **0.993** | 0.17 | 0.173 | 0.276 | 0.258 | 392.79 |
| **Ours-I2V** | **0.509** | **0.689** | **0.993** | **0.58** | **0.144** | **0.269** | **0.182** | **180.81** |

# 4 EXPERIMENTS

## 4.1 IMPLEMENTATION DETAILS

**Dataset.** Our model is trained and validated on the RealEstate10K dataset (Zhou et al., 2018), which contains approximately 10 million frames with associated camera poses, extracted from 80,000 video clips spanning about 10,000 YouTube videos. Each clip provides a trajectory of camera poses that specify both position and orientation along the viewing path using the algorithm COLMAP (Schonberger & Frahm, 2016). Following prior works (He et al., 2025a; Wang et al., 2024b), we evaluate on 100 randomly selected samples from the official test set, which we denote as *RealEstate100*, with sample IDs provided in Appendix J. Furthermore, we evaluate the performance on two dynamic and outdoor datasets, MultiCamVideo (Bai et al., 2025) and DrivingDoJo (Wang et al., 2024a). We also take 100 random examples from each dataset as the test sets, referred to as MultiCamVideo100 and DrivingDoJo100.

**Evaluation metrics.** We evaluate our model and baselines using two groups of metrics. For frame-level quality, we report *LPIPS* (Zhang et al., 2018). For video-level quality, we use *Fréchet Video Distance (FVD)* (Unterthiner et al., 2018) and VBench (Huang et al., 2024), the latter offering targeted evaluation with custom prompts and metrics. We focus on dimensions most relevant to video generation, including *Aesthetic Quality, Imaging Quality, Motion Smoothness, and Dynamic Degree*. To further assess camera motion, we employ the *TransErr* and *RotErr* (He et al., 2025a) metrics. Due to page limitations, we demonstrate more implementation details in Appendix D.

**Baselines.** We mainly compare against four camera-control baselines: MotionCtrl (Wang et al., 2024b), CameraCtrl (He et al., 2025a), and AC3D (Bahmani et al., 2025). Noticed that MotionCtrl, CameraCtrl, and our method support both image-to-video and text-to-video generation. Moreover, we also evaluate the warped-noise-based Go-with-the-Flow (GWTF) (Burgert et al., 2025) method and 3D-based GEN3C (Ren et al., 2025) method.

Table 2: Zero-shot I2V comparisons with MotionCtrl (Wang et al., 2024b), CameraCtrl (He et al., 2025a), Go-with-the-Flow (GWTF) (Burgert et al., 2025), and GEN3C (Ren et al., 2025) methods on the MultiCamVideo100 (upper block) and DrivingDojo100 (lower block) test sets.

| Methods | Aesthetic Quality ↑ | Imaging Quality ↑ | Motion Smoothness | Dynamic Degree | TransErr ↓ | RotErr ↓ | LPIPS ↓ | FVD ↓ |
|---|---|---|---|---|---|---|---|---|
| MotionCtrl-I2V | 0.535 | **0.707** | 0.991 | 0.02 | 0.287 | 0.556 | 0.558 | 908.01 |
| CameraCtrl-I2V | 0.571 | 0.537 | 0.992 | 0.02 | 0.242 | 0.30 | 0.561 | 929.44 |
| GEN3C | 0.546 | 0.571 | 0.988 | 0.13 | 0.212 | 0.343 | 0.521 | 724.74 |
| GWTF | 0.592 | 0.594 | 0.987 | **0.26** | **0.187** | 0.235 | 0.566 | 667.68 |
| **Ours** | **0.601** | 0.623 | **0.993** | 0.25 | 0.191 | **0.230** | **0.298** | **362.57** |
| MotionCtrl-I2V | 0.428 | **0.575** | 0.991 | 0.22 | 1.445 | 0.331 | 0.505 | 672.90 |
| CameraCtrl-I2V | 0.448 | 0.559 | **0.994** | 0.04 | 2.140 | **0.210** | 0.493 | 667.64 |
| GEN3C | 0.440 | 0.521 | 0.985 | 0.45 | 0.937 | 0.320 | 0.482 | 374.83 |
| GWTF | 0.458 | 0.499 | 0.986 | **0.88** | 0.776 | 0.359 | 0.496 | 341.29 |
| **Ours** | **0.461** | 0.536 | 0.984 | **0.88** | 0.397 | 0.265 | **0.158** | **242.67** |

## 4.2 COMPARISONS WITH OTHER METHODS

**Quantitative comparison.** Table 1 presents model performance on Text-to-Video (T2V) and Image-to-Video (I2V) tasks using the RealEstate100 test set, comparing our method with four baselines. "Model"-T2V refers to inference conditioned solely on text, while "Model"-I2V refers to inference conditioned on an image or both image and text. Our model achieves the best overall performance in both video quality and camera control. In terms of video quality, it demonstrates strong results in image quality, temporal consistency, and dynamism. Notably, for I2V, our method increases the dynamic degree by 75.8% over the previous best approach, while maintaining motion smoothness. Our model also achieves state-of-the-art performance in camera motion evaluation compared to existing methods. Furthermore, Table 2 reports the zero-shot I2V results on the MultiCamVideo100 and DrivingDojo100 test sets. We mark the optimal results in bold and the sub-optimal results with an underline. Since these models are trained on different datasets, the zero-shot experiments provide a more objective assessment of the models' generalization ability. The results show that our method also achieves the best or highly competitive performance on these dynamic-scene datasets. This improvement results from the proposed CameraNoise, which directly controls camera motion at the noise level. Combined with our camera pose warping algorithm, it effectively mitigates noise and non-expressive artifacts introduced by numerical control methods. Consequently, our camera-control approach not only preserves video quality but also enables precise manipulation of camera motion, offering new directions for camera-controllable video generation.

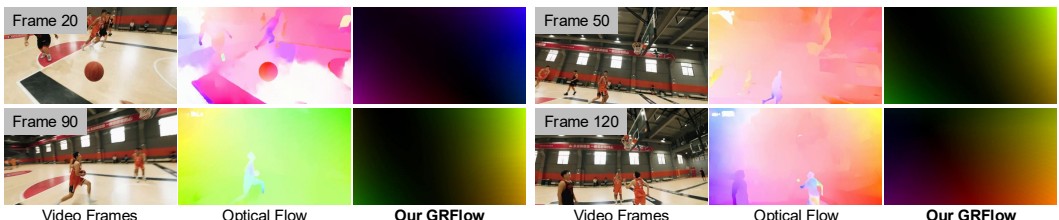

Figure 3: Qualitative comparison between optical flow (Teed & Deng, 2020) and our proposed GRFlow in a real dynamic scenario. GRFlow is designed to capture camera motion information from video frames, while optical flow struggles to separate camera motion from scene appearance. This design ensures that GRFlow does not introduce appearance priors during CameraNoise warping.

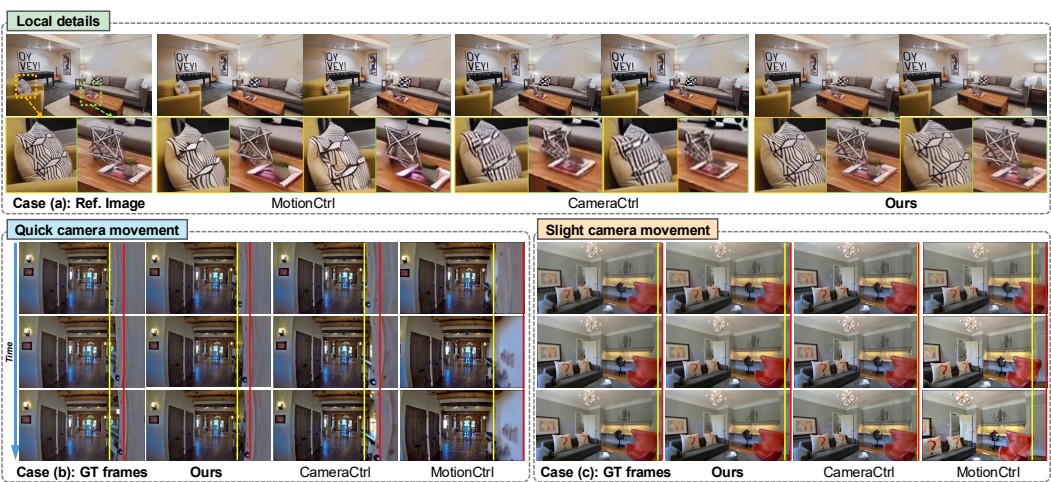

Figure 4: Qualitative comparison on RealEstate10K with MotionCtrl (Wang et al., 2024b) and CameraCtrl (He et al., 2025a). We illustrate two aspects: **1**) visual quality and local details for case (a); **2**) camera movement under fast and slight motion in cases (b) and (c), where vertical **red** and **yellow** lines mark temporal anchor points. Their gap reflects motion magnitude relative to the ground truth.

**Qualitative comparison.** Fig. 3 compares the results of GRFlow with optical flow in video processing. Fig. 4 presents a visual comparison with prior approaches using data sampled from RealEstate10K. Our model significantly outperforms existing methods in preserving fine details

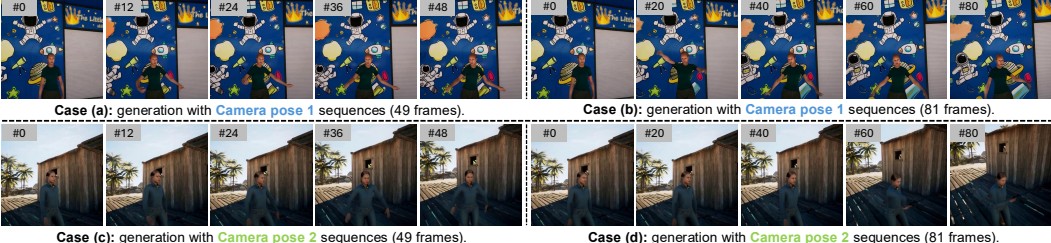

Figure 5: Dynamic results across multiple scenes under different camera poses. In each scene, an anchor point is highlighted with a red circle, and dashed lines indicate the camera's motion trajectory.

within frames and accurately representing camera motion, including rapid and slight camera movement. Besides, in Fig. 5, we give four dynamic and outdoor cases under two kinds of camera poses. User study and more qualitative comparison results can be found in Appendix E and G.

**Case (a):** generation with Camera pose 1 sequences (49 frames).    **Case (b):** generation with Camera pose 1 sequences (81 frames).

**Case (c):** generation with Camera pose 2 sequences (49 frames).    **Case (d):** generation with Camera pose 2 sequences (81 frames).

Figure 6: Stable results for longer videos on MultiCamVideo. We use the same camera sequences (camera poses 1 and 2) to generate videos of 49 (cases a and c) and 81 (cases b and d) frames, respectively. Cases (a-b) and cases (c-d) correspond to results generated from the same input images.

**Comparison of longer video results.** Fig. 6 presents the results of videos with different lengths under the same scenes, and our method maintains stable and coherent temporal consistency. Since the CameraNoise introduced during training does not alter the base model architecture, the trained model retains the longer video synthesis capability provided by 3D RoPE. Consequently, a model trained on 49 frames can directly generate 81-frame videos (typical values for DiT-based models).

## 4.3 ABLATION STUDY

Table 3: Effectiveness of the quality of GRFlow with smoothing algorithm for video generation. We use setting with $\alpha = 1.0$ as the baseline (in gray), which corresponds to results without smoothing.

| Methods | Aesthetic Quality ↑ | Imaging Quality ↑ | Motion Smoothness | Dynamic Degree | TransErr ↓ | RotErr ↓ | FVD ↓ |
|---|---|---|---|---|---|---|---|
| $\alpha = 1.0$ | 0.517 | 0.707 | 0.994 | 0.51 | 0.145 | 0.281 | 181.63 |
| $\alpha = 0.8$ | -0.19% | +0.52% | +0.0% | +1.96% | +0.54% | -8.71% | +7.86% |
| $\alpha = 0.6$ | -0.19% | +0.29% | +0.0% | -3.92% | +3.92% | +2.28% | +1.17% |
| $\alpha = 0.4$ | -0.19% | -0.14% | +0.0% | -7.84% | +0.69% | +4.15% | +0.77% |
| $\alpha = 0.2$ | -15.47% | -2.57% | -0.10% | +13.73% | +0.69% | +4.15% | +5.66% |
| $\alpha = 0.0$ | -13.54% | +1.84% | +0.10% | -37.25% | -376.19% | -211.72% | -54.61% |

**Impact of GRFlow quality on video results.** In Table 3, we show how different $\alpha$ values in the $\Delta E$ smoothing algorithm affect the quality of GRFlow and, consequently, the video generation results. Notably, all quantitative ablation studies are on the RealEstate100 test set for I2V task. The results show that decreasing $\alpha$ (*i.e.,* strengthening the smoothing effect) leads to a notable improvement in generation quality. However, while $\alpha$ can be set arbitrarily close to 0, it cannot be exactly 0, as this would disrupt the numerical structure of $\Delta E$ and cause a sharp degradation in performance.

**Effect of $\alpha$ value in $\Delta E$ smoothing algorithm on GRFlow.** During the generation of GRFlow, the smoothing algorithm's $\alpha$ parameter plays a critical role in shaping the final results. As shown

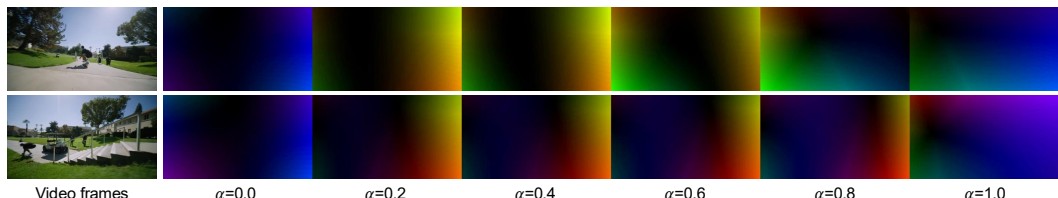

| Video frames | α=0.0 | α=0.2 | α=0.4 | α=0.6 | α=0.8 | α=1.0 |

Figure 7: Visualization of GRFlow under $\Delta\mathbf{E}$ smoothing algorithm with different $\alpha$ values.

in Fig. 7, varying $\alpha$ lead to distinct outcomes, *i.e.,* the changing of GRFlow with different values, highlighting its influence under real-world motion scenarios. This demonstrates that careful tuning of $\alpha$ is essential for achieving stable and accurate camera motion representation.

Table 4: Effectiveness of the weight $\lambda$, which controls the mixing ratio between Gaussian noise and CameraNoise during inference. We use $\lambda = 0$ as the baseline (in gray), where only CameraNoise serves as the initial noise, and gradually increase $\lambda$ to raise the proportion of Gaussian noise.

| Methods | Aesthetic Quality ↑ | Imaging Quality ↑ | Motion Smoothness | Dynamic Degree | TransErr ↓ | RotErr ↓ | FVD ↓ |
|---|---|---|---|---|---|---|---|
| $\lambda = 0.0$ | 0.509 | 0.689 | 0.993 | 0.58 | 0.153 | 0.248 | 196.83 |
| $\lambda = 0.1$ | +0.79% | +1.60% | +0.10% | -5.17% | -0.85% | -5.96% | +0.57% |
| $\lambda = 0.2$ | +0.79% | +1.89% | +0.0% | -5.17% | +5.62% | -8.53% | +12.95% |
| $\lambda = 0.4$ | +1.57% | +1.89% | +0.10% | -3.44% | -6.70% | -17.48% | +6.95% |
| $\lambda = 0.6$ | +1.38% | +1.74% | +0.0% | -8.62% | -45.15% | -57.04% | -5.67% |
| $\lambda = 0.8$ | +1.57% | +4.21% | +0.0% | -6.90% | -135.52% | -123.38% | -30.55% |
| $\lambda = 1.0$ | +2.36% | +4.79% | +0.0% | -12.07% | -145.30% | -121.16% | -22.04% |

**Impact of the fusion ratio $\lambda$ on video results.** Table 4 reports inference performance under different value $\lambda$ in Eq.10, which controls the mixing ratio between CameraNoise and Gaussian noise. As the ratio increases, we observe slight improvements in aesthetic quality and imaging quality, but substantial drops in video dynamism and camera-control accuracy. From the table, a clear trend can be observed: as the proportion of CameraNoise decreases, the effectiveness of camera control drops significantly. Specifically, when $\lambda$ is 0.6, the camera control performance declines by nearly 50%. This result not only quantifies the impact of noise composition on generation but also strongly validates the effectiveness of our approach for controlling camera trajectories in the noise space.

Table 5: Impact of the dynamic scaling training (DST) strategy. $\eta^*$ denotes a fixed value.

| Methods | Aesthetic Quality ↑ | Imaging Quality ↑ | Motion Smoothness | Dynamic Degree | TransErr ↓ | RotErr ↓ | FVD ↓ |
|---|---|---|---|---|---|---|---|
| Training w/o DST | 0.481 | 0.630 | 0.988 | 0.52 | 0.167 | 0.304 | 198.04 |
| DST w/ $\eta^* = 0.9$ | **0.517** | 0.675 | **0.994** | 0.51 | 0.155 | 0.277 | 188.36 |
| DST w/ $\eta \in (0.8, 1.2)$ | 0.497 | 0.642 | 0.993 | 0.55 | 0.160 | 0.270 | 182.95 |
| DST w/ $\eta \in (0.9, 1.1)$ | 0.509 | **0.689** | 0.993 | **0.58** | **0.144** | **0.269** | **180.81** |

**Effect of the dynamic scaling strategy during training.** We introduce camera control signals into the diffusion process via CameraNoise and propose a dynamic scaling training (DST) strategy to adaptively adjust the scaling of rotation matrices in camera extrinsics during training. As shown in Table 5, we conduct a control experiment without the DST strategy and systematically evaluate the effect of varying parameter $\eta$ within the DST strategy. The results show that the adjustments of DST can effectively enhance the model's robustness during inference.

## 5 CONCLUSION

In this work, we introduced CameraNoise, a stochastic representation that embeds camera poses into the noise space independently of scene appearance, enabling temporally coherent video generation. Leveraging a Geometry-guided Reprojection Flow (GRFlow) and a CameraNoise warping algorithm, our method preserved the Gaussian prior of the diffusion process and maintains consistent noise propagation under camera transformations. Integrating CameraNoise into the video diffusion enables precise camera control, producing high-quality and stable videos. Experimental results showed that our approach achieved higher video quality and stronger visual performance than prior methods, effectively addressing the challenge of imprecise camera control.

ETHICS STATEMENT

Our work introduces CameraNoise, a stochastic representation for embedding camera poses into the noise space, enabling temporally coherent and camera-controllable video generation. While our method can benefit applications such as animation and virtual production, it could also be misused to generate realistic synthetic videos, potentially raising concerns regarding privacy, consent, and misinformation.

To mitigate these risks, all experiments are conducted using publicly available datasets with appropriate licenses, and no personally identifiable sensitive information is included. We encourage responsible use of our models and advise careful evaluation of ethical and societal implications when deploying such generative technologies. Our primary goal is to advance scientific research in controllable video generation, and we advocate for precautions to prevent malicious or unethical applications.

USE OF LARGE LANGUAGE MODELS STATEMENT

We did not use any large language models (LLMs) in the development, experimentation, or writing of this work. All models, algorithms, and analyses presented in this paper were implemented and evaluated without reliance on LLMs. The datasets, training procedures, and evaluation protocols are entirely independent of LLM-generated content. This ensures that our results and conclusions are fully derived from the methods described in this paper and are not influenced by any external LLM outputs.

REPRODUCIBILITY STATEMENT

We provide a detailed description of our CameraNoise framework, including the Geometry-guided Reprojection Flow and CameraNoise warping algorithm. All datasets used in our experiments are publicly available with proper licenses. We include comprehensive training details, hyperparameter settings, and evaluation protocols in the paper and supplementary material to facilitate replication. While the full source code will be made publicly available upon publication, all results reported in this work can be reproduced using the provided descriptions and publicly accessible data.

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

## A  PROOF: CONTINUITY OF GRFLOW.

In this section, we discuss our GRFlow algorithm, providing a formal proof of its continuous properties. We first define the continuous pixel domain $\Omega_c \subset \mathbb{R}^2$ and let the camera pose vary continuously in time $t \in [o, T]$: $\mathbf{E}(t) \in \mathrm{SE}(3)$. Then, the continuous flow is:

$$\Phi : \Omega_c \times [0, T] \to \Omega_c, \quad \Phi(x, y, t) = \pi(\mathbf{E}(t) \cdot \ell(x, y)), \tag{13}$$

where $\ell$ is the lifting map and $\pi$ is projection to 2D. In practice, we sample pixels on a discrete grid $\Omega \subset \Omega_c$ and time is discretized into frame $t_i$. So, we can compute the GRFlow as:

$$\mathcal{G}_r^{i \to i+1}(x, y) = \pi(\Delta\mathbf{E}_i \cdot \ell(x, y)), \quad (x, y) \in \Omega, \tag{14}$$

with $\Delta\mathbf{E}_i = \mathbf{E}_i^\top \times \mathbf{E}_{i+1}$. As pixel resolution increases $D \to \infty$ and frame interval $\Delta t \to 0$, the discrete flow $\mathcal{G}_r$ converges pointwise to the continuous map $\Phi$.

$$\lim_{|\Omega| \to \infty, \Delta t \to 0} \mathcal{G}_r^{i \to i+1}(x, y) = \Phi(x, y, t_i), \quad \forall (x, y) \in \Omega_c. \tag{15}$$

Because we smooth the $\Delta\mathbf{E}$ during reprojection, the map is smooth. So, $\mathcal{G}_r$ is effectively a finite-sample approximation of a continuous flow on the 2D manifold, *i.e.*, $\mathcal{G}_r \approx \Phi|_\Omega$. Furthermore, since continuous coordinates are sampled on the discrete grid $\Omega$, sub-pixel displacements are typically approximated or interpolated, introducing minor errors. However, in our experiments, we find these errors to be negligible and they do not affect the results. Time Complexity of $\mathcal{G}_r$: furthermore, to construct GRFlow, we initialize a $D \times D$ grid. For each pixel in the grid, we compute $d \times \mathbf{K}$, where $\mathbf{K} \in \mathbb{R}^{3 \times 3}$ is the intrinsic matrix, resulting in a computational cost of $\mathcal{O}(D^2)$. The subsequent transformation using $\Delta\mathbf{E}$ also incurs $\mathcal{O}(D^2)$ operations. Hence, the overall time complexity of our algorithm is $\mathcal{O}(D^2)$ per frame.

## B  CAMERANOISE: ONE-TO-ONE MAPPING FROM CAMERA POSE VIA GRFLOW.

### B.1  WHY DIRECT NOISE INTERPOLATION DISRUPTS THE GAUSSIAN PRIOR?

In our framework, we consider a noise $N \sim \mathcal{N}(0, \sigma^2)$, where each pixel is independently drawn from a Gaussian distribution with zero mean and variance $\sigma^2$. This independent Gaussian prior is a fundamental assumption in diffusion-based generative models, ensuring that the denoising process operates on ideal white noise. When performing warping of the noise map using GRFlow, the target pixel coordinates often fall between discrete grid points, requiring interpolation (such as bilinear or bicubic) to obtain the warped noise values:

$$N'(x', y') = \sum_{i,j} w_{ij} N(x_i, y_j), \tag{16}$$

where $w_{ij}$ are interpolation weights satisfying $\sum_{i,j} w_{ij} = 1$. However, such interpolation inherently modifies the statistical properties of the original noise. Specifically, the output pixel $N'(x', y')$ is a weighted sum of neighboring noise values, which leads to a variance of:

$$\mathrm{Var}(N') = \sum_{i,j} w_{ij}^2 \sigma^2 < \sigma^2. \tag{17}$$

This variance reduction diminishes the noise amplitude, thereby violating the original Gaussian prior. Moreover, after interpolation, adjacent pixels in the warped noise map receive contributions from overlapping input pixels, which introduces spatial dependencies. Consequently, the original assumption of independence in the Gaussian prior is no longer valid.

## B.2 PROOF: ONE-TO-ONE MAPPING AMONG CAMERA POSE, GRFLOW, AND CAMERANOISE.

Suppose we have the camera parameters $\mathbf{K} \in \mathbb{R}^{3 \times 3}, \mathbf{E} \in \mathrm{SE}(3)$, for a given frame of a video. Under the standard camera model, a 3D world point $X \in \mathbb{R}^3$ projects to a pixel coordinate $x \in \mathbb{R}^2$:

$$x \sim \pi(\mathbf{K}, \mathbf{E}, X) = K[\mathbf{R}|\mathbf{t}]X. \tag{18}$$

Our GRFlow is defined as the displacement field computed from the changes in camera matrices between consecutive frames:

$$\mathcal{G}_r(x) = f(\mathbf{K}_t, \mathbf{E}_t, \mathbf{K}_{t+1}, \mathbf{E}_{t+1}). \tag{19}$$

Meanwhile, the warped CameraNoise $\rho_{t+1}$ corresponds to the advected noise field obtained by solving the PDE:

$$\rho_{t+1} = \mathrm{Advect}(\rho_t, \mathcal{G}_r). \tag{20}$$

To establish that the warped noise and the camera parameters form a one-to-one and unique mapping, *i.e.,* $| \rho_{t+1} \leftrightarrow (\mathbf{K}_{t+1}, \mathbf{E}_{t+1})$, we assume the following conditions:

- Optical invertibility: The camera projection is a single-valued mapping, *i.e.,* different 3D points do not project to the same pixel, and the projection is locally invertible.
- Uniqueness of GRFlow: The displacement field generated from continuous camera parameters uniquely determines the advect vector for each pixel.
- Uniqueness of the PDE: Given initial conditions and a smooth velocity field, the PDE admits a unique solution.

Under these assumptions, the mapping from GRFlow to the PDE advected velocity field is unique. Specifically, for each pixel $x$, the GRFlow induced by the camera parameter transformation is uniquely determined:

$$v(x) = \mathcal{G}_r(x) = \pi^{-1}(\mathbf{K}_t, \mathbf{E}_t, x) - \pi^{-1}(\mathbf{K}_{t+1}, \mathbf{E}_{t+1}, x). \tag{21}$$

Formally, assume there exist two distinct sets of camera parameters $(\mathbf{K}', \mathbf{E}') \neq (\mathbf{K}_{t+1}, \mathbf{E}_{t+1})$ that yield the same GRFlow for every pixel $x$. This contradicts the local invertibility of the camera projection at the pixel level, which ensures that no two different 3D configurations can project identically. Therefore, each GRFlow corresponds to a unique set of camera parameters, establishing a one-to-one correspondence.

Therefore, since the GRFlow is uniquely determined by the camera parameters, and the PDE solution is uniquely determined by the GRFlow and the initial noise, there exists a one-to-one correspondence between CameraNoise and the camera parameters as:

$$(\mathbf{K}_{t+1}, \mathbf{E}_{t+1}) \xrightarrow{\text{GRFlow}} \mathcal{G}_r \xrightarrow{\text{PDE Advect}} \text{CameraNoise}. \tag{22}$$

## C JACOBIAN MATRIX DEFINED IN CAMERANOISE WARPING.

In our CameraNoise warping algorithm, we employ an area scaling factor using the *Jacobian* matrix, which can correct the noise and ensure statistical consistency throughout the warping. We initialize the 2D Gaussian noise and warp it with the GRFlow as: $G(x)' = f(G(x)) = x + \mathcal{G}_r(x); x = (u, v)$, where $f$ denotes the mapping function. Within a local patch, $f(x)$ of pixel coordinates is approximated using a first-order Taylor expansion, yielding:

$$f(x + \Delta x) \approx f(x) + \mathcal{J}(x) \times \Delta x, \mathcal{J}(x) = I + \nabla d(x), \tag{23}$$

where $\nabla d(x)$ represents the *Jacobian* matrix of the mapping function $f(x)$ at $x$. It describes local linear transformation properties such as scaling, rotation, and shearing. To solve the $\mathcal{J}(x)$, we reuse the corresponding camera pose $\mathbf{E}$. We assume that:

$$p = K^{-1}\tilde{x} = \begin{bmatrix} (u - c_x)/f_x \\ (v - c_y)/f_y \\ 1 \end{bmatrix}, \quad X = d \times p, \tag{24}$$

where $d$ denotes the depth. Then, we get $X' = \mathbf{R} \times X + \mathbf{t}$ and project it onto the image plane $(u', v')$ with respect to the original pixel coordinates $(u, v)$ as:

$$u' = f_x \frac{X'_x}{X'_z} + c_x, \quad v' = f_y \frac{X'_y}{X'_z} + c_y. \tag{25}$$

For $\partial u'/\partial u$, using the quotient rule, we have:

$$\frac{\partial u'}{\partial u} = f_x \frac{(\partial_u X'_x)X'_z - X'_x(\partial_u X'_z)}{(X'_z)^2}. \tag{26}$$

The other three partial derivatives $[\partial u'/\partial v; \partial v'/\partial u; \partial v'/\partial u]$ are also computed in a similar manner. These partial derivatives are then combined to form the $\mathcal{J}$. Finally, the local area scaling factor is given by the absolute value of the determinant of $\mathcal{J}$ as $s(x) = |\det\mathcal{J}|$. Expansion occurs when the local volume increases ($\det J(x) > 1$), causing a single source pixel to split into multiple target pixels. In the bipartite graph, this results in one-to-many edges, with weights scaled by the local flow density. Contraction occurs when the local volume decreases ($\det J(x) < 1$), such that multiple source pixels map to a smaller region. In the bipartite graph, this corresponds to many-to-one edges, with missing regions filled using noise from the backward flow.

# D  ADDITIONAL IMPLEMENTATION DETAIL.

**Evaluation metric details.** To assess camera motion, we employ the *TransErr* and *RotErr* (He et al., 2025a) metrics. RotErr measures rotational consistency of the camera using $\mathbf{R}$ matrices, while TransErr quantifies translation error as the $L2$ distance of the translation vectors $\mathbf{t}$:

$$\text{RotErr} = \sum_{j=1}^{m}(\sum_{i=1}^{n} \arccos \frac{tr(\mathbf{R}^i_{gen}\mathbf{R}^{i\mathrm{T}}_{gt})) - 1}{2})/m, \tag{27}$$

$$\text{TransErr} = \sum_{j=1}^{m}(\sum_{j=1}^{n} \|\mathbf{t}^i_{gt} - \mathbf{t}^i_{gen}\|^2_2)/m. \tag{28}$$

For $m$ videos, we compute the average over the $n$ frames of each video, and take the mean across all $m$ videos. To improve evaluation accuracy, we re-estimate camera poses for both ground-truth and generated videos using the advanced camera estimation model VGGT (Wang et al., 2025).

**Implementation details.** Our experimental details are divided into two parts: 1) CameraNoise generation: Experiments were conducted on a single NVIDIA GPU. We used the VGGT model to estimate camera parameters for data without pose annotations and generated GRFlow and CameraNoise using our proposed algorithm. Our method is **real-time capable**, processing videos at approximately 10 frames per second. 2) Integration of CameraNoise into video diffusion models: Experiments were conducted on 32 NVIDIA GPUs for model fine-tuning. We adopted the mainstream DiT Wan 2.1 model (Wan et al., 2025) as our training framework. CameraNoise was injected at the noise level, and the model was trained on the RealEstate10K training set using a LoRA-based training approach. We train videos with 49 frames and $1024 \times 576$ resolution. We set the LoRA rank to 32 and trained the model with a learning rate of 1e-4 and a batch size of 32.

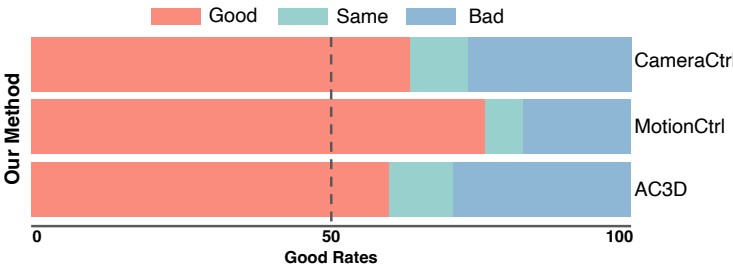

Figure 8: User study on 30 videos. Longer pink bars indicate higher satisfaction with our method.

# E    USER STUDY

Fig. 8 presents the results of our user study evaluating the method with CameraCtrl (He et al., 2025a), MotionCtrl (Wang et al., 2024b), and AC3D (Bahmani et al., 2025) models, from a human perception perspective. In the study, participants were asked to compare our approach with baseline methods in terms of camera control and visual appeal. The results show that participants consistently preferred our method, indicating that it produces outputs that are not only more realistic but also more visually compelling, which also highlights its advantages over existing camera-controllable methods.

Table 6: Comparison of runtime overhead between our method and the Go-with-the-Flow (GWTF) (Burgert et al., 2025). We compare the runtime when processing an 81-frame video. The upper part reports the generation time of optical flow and GRFlow, while the lower part shows the runtime for optical-flow-based warped-noise generation and the runtime of our GRFlow-to-CameraNoise generation.

| Methods | Total time cost (seconds) ↓ | Per frame time cost (seconds) ↓ |
|---|---|---|
| Optical-flow generation (GWTF) | 4.455 | 0.055 |
| **GRFlow generation (Ours)** | **1.476** (×3.02) | **0.018** |
| Optical-flow to Warped-Noise (GWTF) | 9.558 | 0.118 |
| **GRFlow-to-CameraNoise (Ours)** | **6.802** (×1.41) | **0.084** |

# F    TIME COST.

**Time cost of GRFlow and CameraNoise generation.** Table 6 reports the generation time of our GRFlow method and the GRFlow-to-CameraNoise generation. We evaluate our method and the Go-with-the-Flow (GWTF) (Burgert et al., 2025) on an 81-frame video. In the first stage (from camera pose to GRFlow), our approach requires only *0.018 seconds per frame*, whereas optical-flow–based methods are 3.02× slower. In the second stage (from GRFlow to CameraNoise), our synthesis process takes just *0.084 seconds per frame*, achieving a 1.41× speed-up compared to the optical–flow–based approach GWTF. Therefore, our method can achieve a processing speed of *9.8 frames per second* from the initial camera pose to CameraNoise.

**Time cost of mode training and inference.** Our model is trained based on the Wan2.1 model. Since we do not modify the base model architecture, the training and inference times are roughly the same as the base model. On NVIDIA A100 hardware, we train the model using 32 GPUs with a total batch size of 32, a training resolution of 1024×576, and video samples of 49 frames. Each training step takes approximately 55.3 seconds. During inference, generating a 49-frame video at 1024×576 resolution with 25 steps takes about 390 seconds. When the frame count is increased to 81, the inference time rises to 780 seconds.

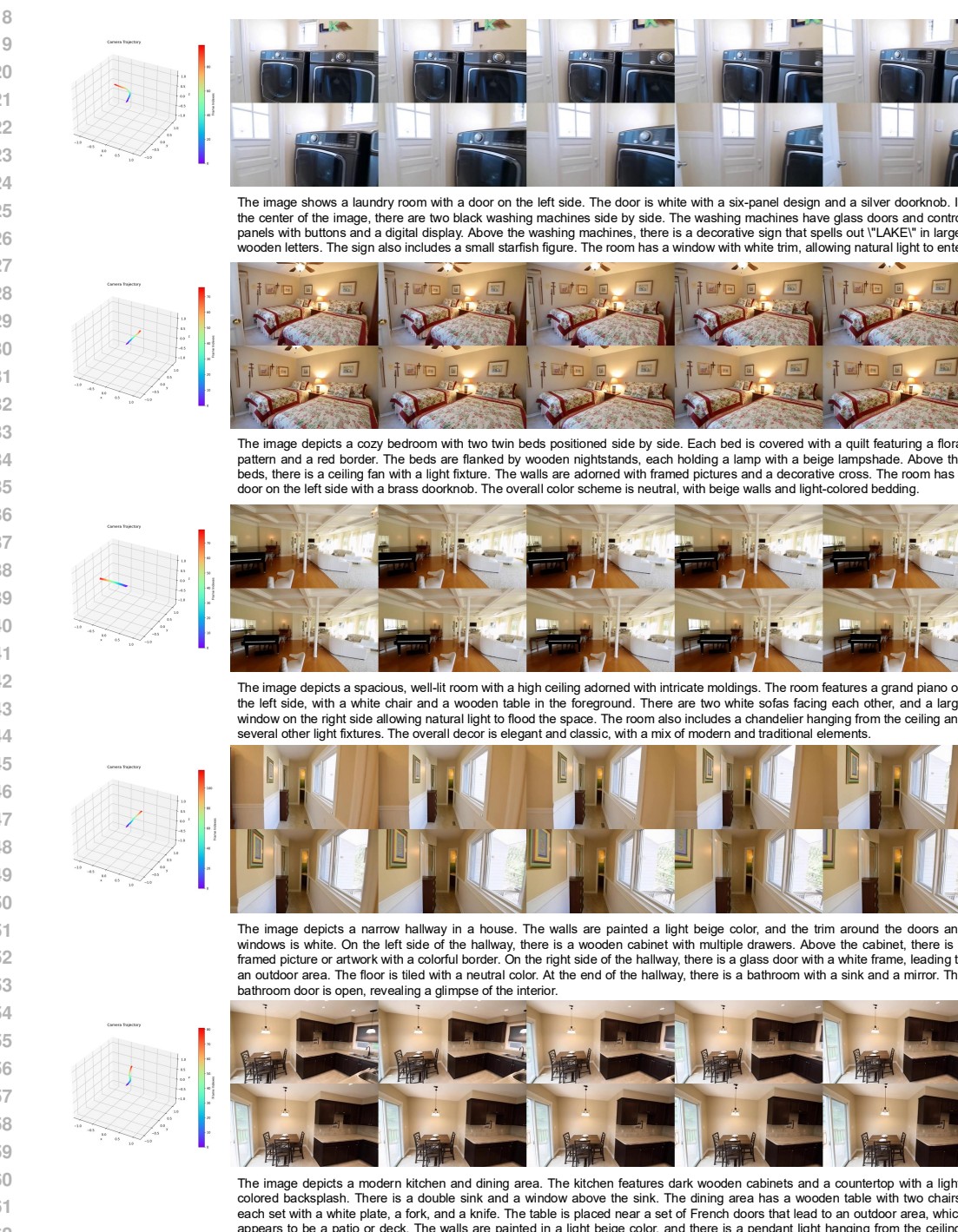

Figure 9: Qualitative results of text-to-video generation (1/2) for video sequences under complex camera poses in RealEstate10K dataset. For each case, the leftmost panel shows the camera trajectory used for control, with prompts provided below each image.

## G  MORE QUALITATIVE COMPARISON.

Additionally, we provide more qualitative comparisons, including text-to-video generation results in Fig. 9 and Fig. 10, and image-to-video generation results in Fig. 11 and Fig. 12. These output videos are all conditioned by the input camera pose via our CameraNoise algorithm.

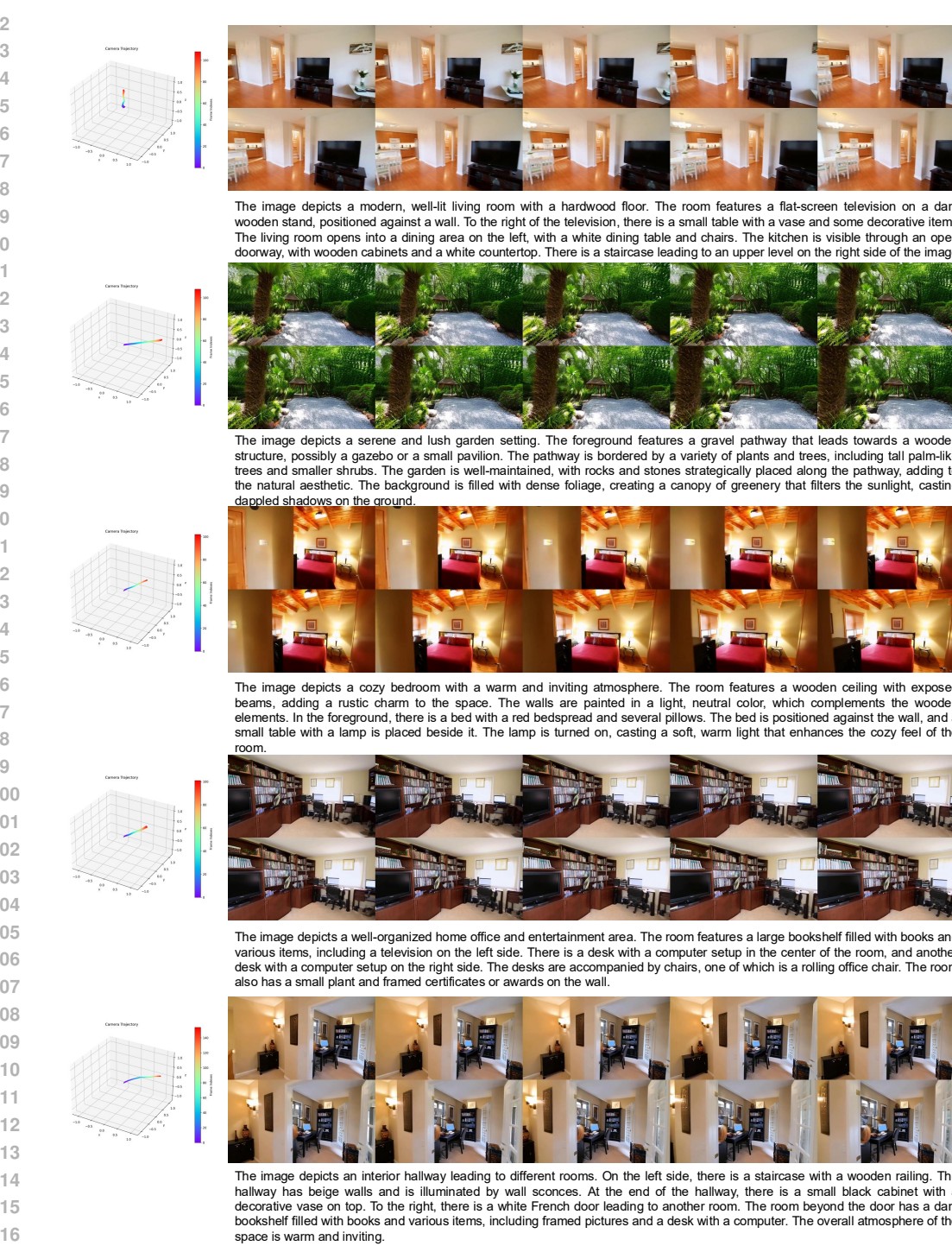

Figure 10: Qualitative results of text-to-video generation (2/2) for video sequences under complex camera poses. For each case, the leftmost panel shows the camera trajectory used for control, with prompts provided below each image.

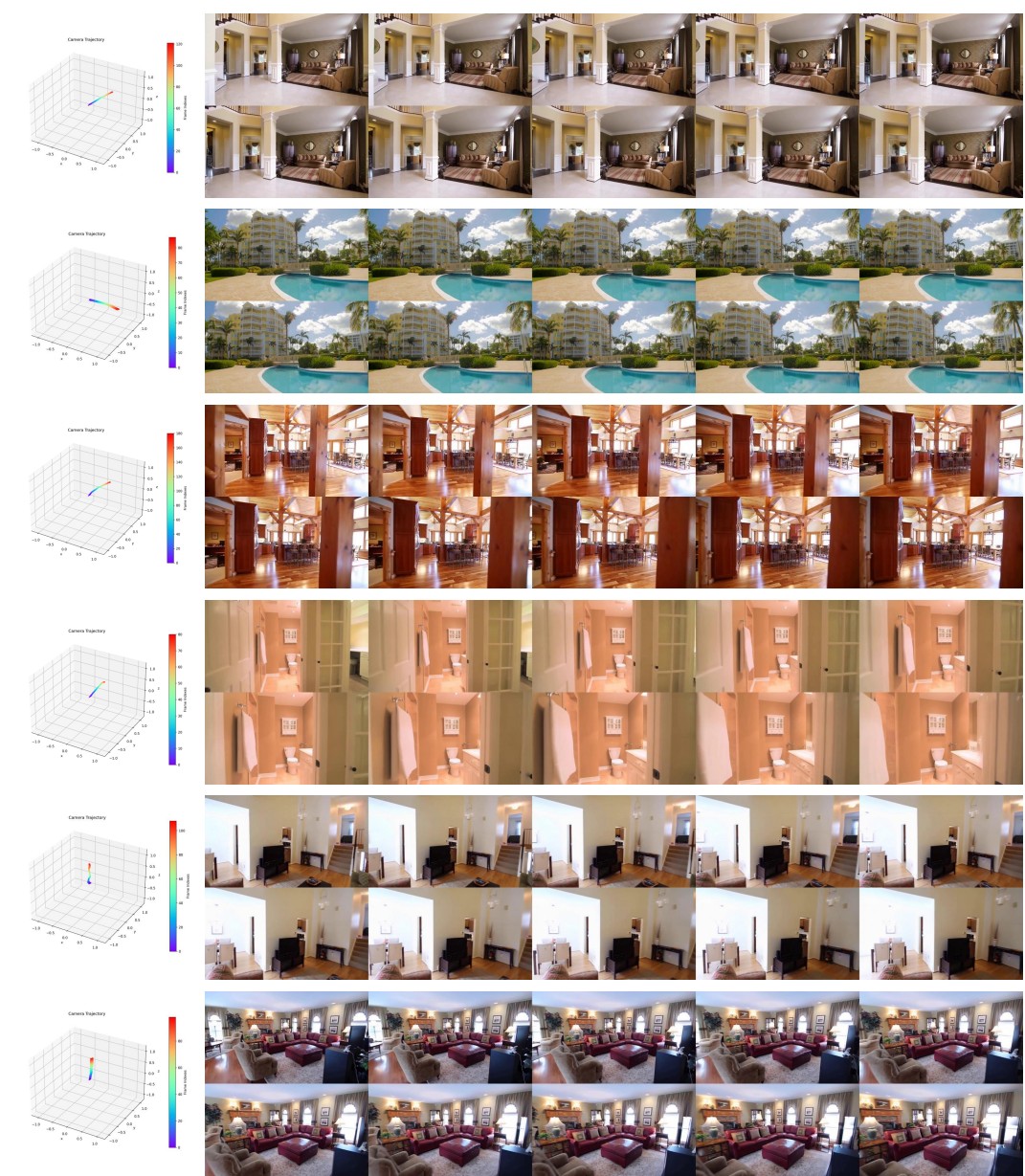

Figure 11: Qualitative results of image-to-video generation (1/2) for video sequences under complex camera poses. For each case, the leftmost panel shows the camera trajectory used for control, and the reference image corresponds to the first frame.

## H  CHALLENGING SCENE.

In common camera scenarios, videos are typically captured from a third-person perspective, as illustrated in datasets such as RealEstate10K, MultiCamVideo, and DrivingDoJo. However, real-world data also contains a large number of *first-person selfie videos*, such as the examples shown in Fig. 13 (a) and (b). We observe that such first-person videos pose greater challenges for camera-control models. In Fig. 13, we present two reasonably good cases and two representative failure cases. The two good cases correspond to situations where the person is holding the camera and moving it upward, and our model is able to predict the camera motion correctly.

In contrast, the failure cases reveal two different issues. In Fig. 13 (a), although the video appears to show a "zoom-out" camera motion, such a movement is physically impossible for a handheld

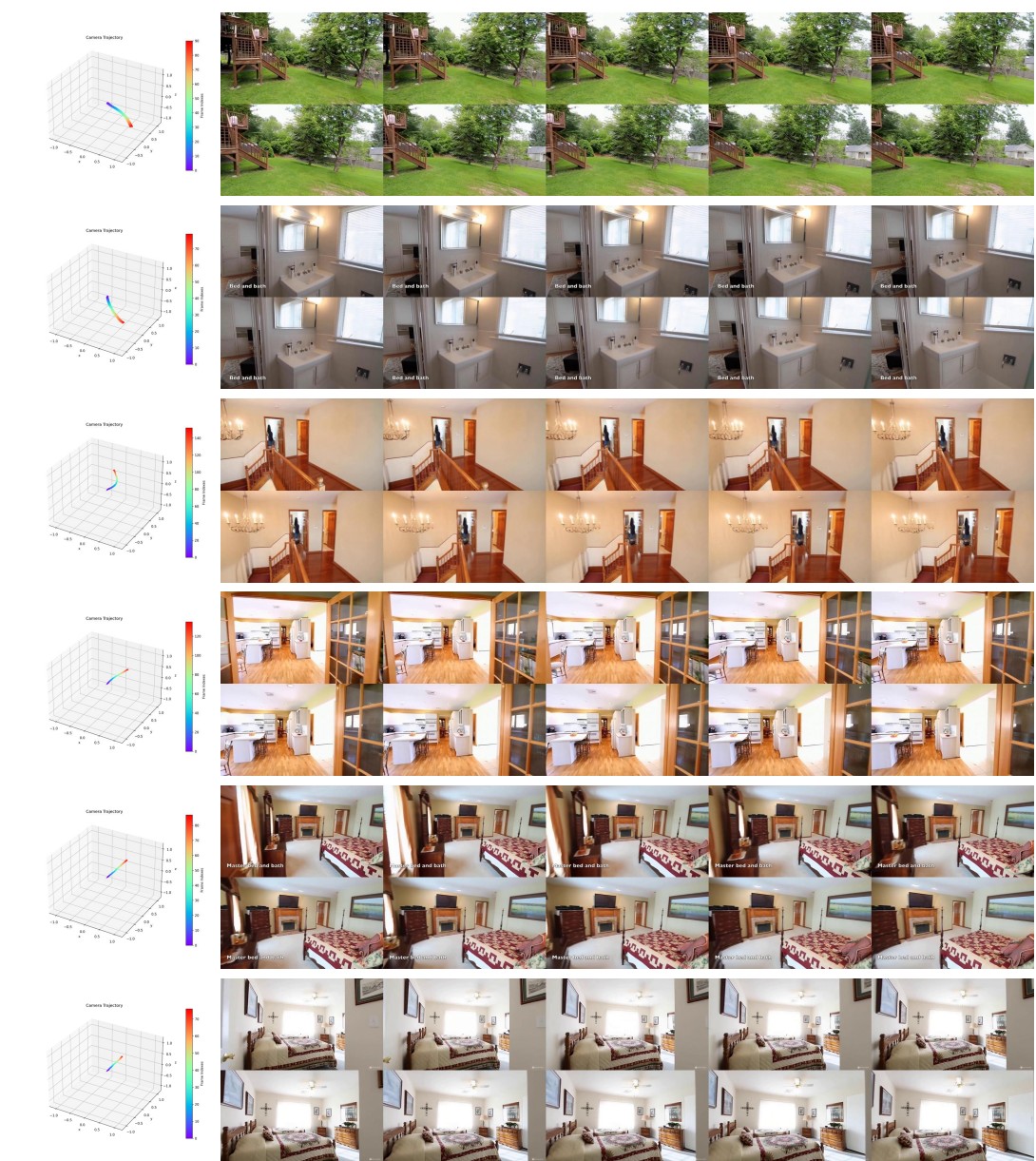

Figure 12: Qualitative results of image-to-video generation (2/2) for video sequences under complex camera poses. For each case, the leftmost panel shows the camera trajectory used for control, and the reference image corresponds to the first frame.

selfie setup due to the fixed length of the selfie stick. However, the model still attempts to synthesize this non-existent camera translation, producing an unrealistic and physically implausible camera motion in the generated result (in red boxes). In Fig. 13 (b), the failure arises because the person has physically shifted the camera's position. Although the person physically moved the selfie camera (to the right), the visual content did not change accordingly (in red boxes) with the camera motion, leading to an inherently inconsistent and unreasonable result.

## I    PERFORMANCES UNDER OUT-OF-DISTRIBUTION (OOD) SCENARIOS.

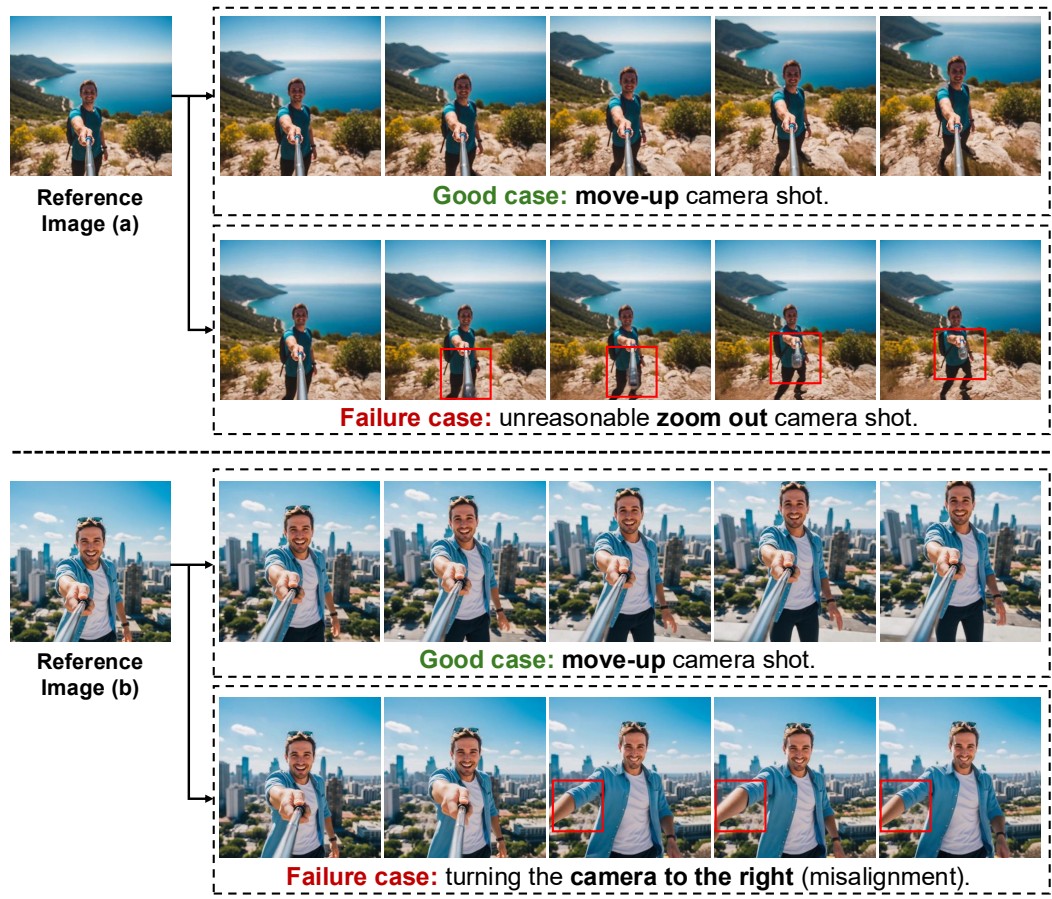

Figure 13: Challenging first-person selfie scenes that pose difficulties for camera-control models.

Table 7: Quantitative analysis of CameraCtrl (He et al., 2025a), MotionCtrl (Wang et al., 2024b), Go-with-the-Flow (Burgert et al., 2025), and GEN3C (Ren et al., 2025) in dynamic OOD scenarios with image-to-video generation. Since there are no ground-truth videos available for these OOD scenarios, we evaluate the models using video quality metrics provided by VBench (Huang et al., 2024). The optimal results are in bold, while the sub-optimal results are marked with underlining.

| Methods | Aesthetic Quality ↑ | Imaging Quality ↑ | Motion Smoothness | Dynamic Degree |
|---------|---------------------|-------------------|-------------------|----------------|
| CameraCtrl | 0.635 | 0.578 | 0.991 | 0.0 |
| MotionCtrl | 0.633 | 0.695 | 0.987 | 0.0 |
| Go-with-the-Flow | 0.636 | 0.597 | 0.986 | **0.33** |
| GEN3C | 0.652 | 0.651 | **0.992** | 0.17 |
| **CameraNoise** | **0.712** | **0.707** | 0.991 | 0.25 |

Out-of-Distribution (OOD) scenarios are commonly used as important data to evaluate model robustness. To this end, we evaluate the performance of MotionCtrl (Wang et al., 2024b), CameraCtrl (He et al., 2025a), Go-with-the-Flow (Burgert et al., 2025), GEN3C (Ren et al., 2025), and our method across six types of scenes: valleys, fields, lakes, deserts, forests, and amusement parks. For these scenes, we apply six typical camera motions, including move-up, move-down, move-left, move-right, move-clockwise, and zoom-in shots, which are also the typical camera movements specified in the GEN3C method. Table 7 presents the quantitative analysis of these methods across the scenes, and Fig. 14 visualizes four selected scenarios.

From these OOD quantitative and qualitative results, we summarize the characteristics and limitations of the referred mainstream methods under OOD settings: 1) MotionCtrl and CameraCtrl exhibit significant deviations in camera following, indicating weak robustness in camera control.

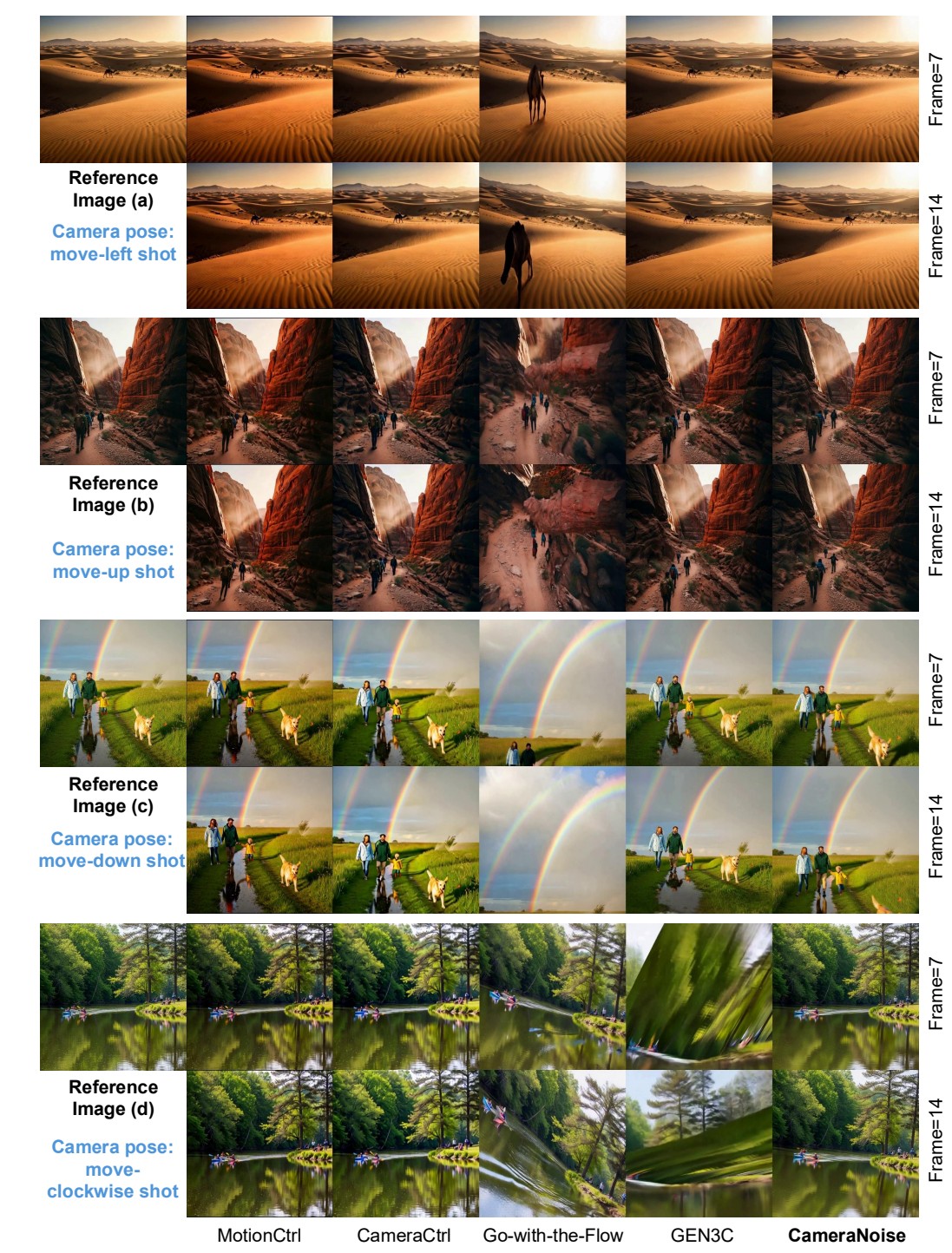

Figure 14: Qualitative results of CameraCtrl (He et al., 2025a), MotionCtrl (Wang et al., 2024b), Go-with-the-Flow (Burgert et al., 2025), and GEN3C (Ren et al., 2025) in dynamic OOD scenarios with image-to-video generation.

2) Go-with-the-Flow often exhibits significant camera motion deviations and can sometimes result in collapsed or distorted video content. 3) GEN3C generates almost entirely static scenes where objects cannot move, resulting in rigid video content. Additionally, due to its reliance on 3D feature modeling, it is prone to scene penetration issues. In contrast, our method demonstrates superior per-

formance in OOD scenarios, excelling in camera control accuracy, content plausibility, and motion dynamics.

## J  EVALUATION DATA IN REALESTATE10K.

The IDs of the RealEstate10K test data used in our experiments are listed as follows:

| | | | |
|---|---|---|---|
| '0e512d350465a63c', | '55c99f9550523caa', | '6da3a9910fddd89e', | 'dd1c1d26525a2a1b', |
| 'e7020d449f50c737', | 'b3944d7eaf4f122b', | 'b5cc912bdce6fabd', | 'b859c7a7329c17b7', |
| 'b16d261e94f32108', | '82a317cb88729fe1', | '021575237abe0684', | '4881a65d7476d6dd', |
| '79bf72c63958d26a', | 'deb368fb90770550', | '1bf668db0194cf83', | '33f1be3a9ccf4e4b', |
| 'db2a88aeac0528ef', | '7526c18f191bcc1a', | '9d3b223e43672fb9', | '58c701594649d4e3', |
| 'c45cab04c3b22166', | '560e521aa9e864da', | 'bc70abfcfd247074', | '4242fb49c775710c', |
| '3b7443b24830d388', | '00cf0a94235771bb', | 'bb181e68415b169e', | '3e3d858083d20eab', |
| '0a7c052273895bb3', | 'bd3728fa823e6eb9', | '9b765910ee6573ec', | '446626a2bd617d24', |
| 'd5c6ad22b14eccef', | '7fad45df233ddce8', | '94e35563d865a2d6', | 'c35d057c102fe5ae', |
| '4daf919100e878ac', | '6986840ead0c9e9d', | '55e902a2cd2e976a', | '114d9c301b847239', |
| 'd35f508ebd80e610', | 'b8fda11b15ac85ff', | '91234df26c87a72b', | '86e00a902518e491', |
| '3018aa8ad3eb5dca', | 'bb6fa5bdafc14e8c', | '5d9f7f0205f7bee5', | '511ad5dc10db6932', |
| '6f243139ca86b4e5', | 'dd5288bacc7da7cf', | '23099812f662b3ec', | '188e6f96fa74ebe7', |
| '3fb3327a177a0175', | '46e0654ccb5d88cf', | '1b09ad5460c05077', | 'bf756257ffdd0017', |
| 'b43aa92e530e2aa5', | '14cf1f92ca13d605', | '122cb7d5ea4a99df', | '6beedb01303bb667', |
| '1b2937e192040745', | 'a3bc75f0a32b1501', | '8317ac8848eb60de', | '645cc7949386d427', |
| '861e80a1959788a7', | 'b2fa3530dcd34093', | '93a8bf0ecd7eafcf', | 'c99afff025000694', |
| '2f7f2369486cc959', | '6ee670df48229b4e', | 'f1d9d9caf8269fe6', | '330b925cef643b3f', |
| '5151d3969e328df1', | '5a15212752d1659a', | 'c8c2d887b38f6dc5', | '3e68931874661724', |
| 'eac593ae8fff36a3', | '7c7bc5285126e6ad', | '504e7ca0f7bb3427', | 'a6bc234b1ce6ca2b', |
| '8f04f919b046336c', | '4c69bf407b142b93', | 'a1d38185b8f59a4b', | 'a8baf24bbae943a0', |
| '0ed8b86b87a30d38', | '5bb9c1498799204b', | '02b59cd60efb924e', | 'd6b3d15ff42247fb' |
| '83ceef672f798063', | 'ce8bc5948d0dd3f3', | 'edbdb6ad0b956efb', | |
| 'a552d52c34c2c920', | 'c3c33ceed1308b42', | '0463d74358aca878', | |
| '0e8a52a174610350', | '0c916bcc9351521e', | '56ae4fff81255579', | |
| '4f0aa4ef8976dc1b', | 'ba604e2b0ab1be25', | '40f92f1e65a5e1dd', | |

