# OpenReview forum: "CameraNoise: Learning Precise Camera Control with Video Diffusion in Noise Space"
_ICLR.cc/2026/Conference — Submitted to ICLR 2026_

### Official Review · Reviewer_c7T5 · 2025-10-16

**Soundness:** 3
**Presentation:** 3
**Contribution:** 3
**Rating:** 4
**Confidence:** 5

**Summary:**

This paper proposes CameraNoise, a method for camera-controlled video diffusion. CameraNoise embeds camera poses directly into the diffusion noise instead of using a conditioning signal within the model blocks, e.g., based on Plucker coordinates. For this, the paper proposes Geometry-guided Reprojection Flow (GRFlow), an alternative to using optical flow for noise warping, where the main motivation is to disentangle camera motion from visual content.

**Strengths:**

- Disentangled camera motion and visual content: Constructing warped noise where the camera motion is disentangled from the visual content is sound and a good idea compared to previous optical flow-based methods.

**Weaknesses:**

- Missing comparisons: Previous works such as Go-with-the-Flow [1] also use noise warping for camera-controlled video modelling. However, there are no comparisons with that line of work. Moreover, there is no comparison with GEN3C [2], a method that uses depth-based warping for 3D inductive bias.
- Lack of video results: The supplementary website is good but it is missing some more videos. First, it would be great to show generalization beyond RealEstate10K. So ideally the paper can show multiple OOD videos with the same trajectory to show that the model is stable across scenes and consistent w.r.t. camera control precision. Moreover, side-by-side comparisons with baselines would be good. Lastly, it would be great to show if noise from an input video can be transferred to other video generations.

[1] Burgert et al., Go-with-the-Flow: Motion-Controllable Video Diffusion Models Using Real-Time Warped Noise, CVPR 2025 \
[2] Ren et al., GEN3C: 3D-Informed World-Consistent Video Generation with Precise Camera Control, CVPR 2025

**Questions:**

The paper seems sounds and I like the method. However, there is some lack of comparisons and visual results.
I would like authors to address following questions:
- How does the method compare to recent methods such as Go-with-the-Flow or GEN3C?
- How about some additional visual results such as results on scenes with dynamic objects, and comparisons with previous works?

I currently rate the work below the acceptance threshold but would be happy to consider my rating depending on the rebuttal.

---

> ### Author Response · Authors · 2025-11-20
>
> We thank reviewer c7T5 for the thoughtful and constructive comments. It is evident that the reviewer spent considerable time carefully reading our work and provided valuable insights and suggestions. We really appreciate the detailed feedback and the recognition of the strengths in our approach. Below, we address the specific points raised by the reviewer.
>
> > Question: Missing comparisons with Go-with-the-Flow and GEN3C.
>
> **Response:** We thank the reviewer for pointing this out! We conduct additional comparison experiments on the MultiCamVideo and DrivingDoJo datasets against Go-with-the-Flow and GEN3C to demonstrate the superiority of our model. These results have been updated in our manuscript, and detailed metric comparisons can be found *in Table 2 (highlighted in blue) at Line 364* of the revised manuscript.
> The results show that, even though model like Go-with-the-Flow is trained on 4M samples, our model still outperforms or matches existing methods in VBench, FVD, LPIPS, and camera control accuracy. We attribute this to our method, which introduces a novel appearance-decoupled camera motion representation called CameraNoise and effectively avoids conflicts between noise priors from methods like optical flow and the visual content. We really appreciate your insightful suggestion!
>
> > Question: Lack of more video results, including multiple OOD videos with the same trajectory to show that the model is stable across scenes and consistent w.r.t. camera control precision. Moreover, side-by-side comparisons with baselines would be good. Lastly, it would be great to show if noise from an input video can be transferred to other video generations.
>
> **Response:** We thank the reviewer for the suggestion! We have added additional dynamic scene examples to further demonstrate the effectiveness of our method. These video examples have been updated on the supplementary website and include:
> - OOD videos with the same trajectory in “Rebuttal-A”;
> - Stable cross-scene cases in “Rebuttal-B”;
> - Side-by-side comparisons with baselines in “Rebuttal-E”;
> - Cases transferring the noise from an input video to other video generations in “Rebuttal-D”.
>
> You can find the parts of “Rebuttal-A, B, D, and E” in our anonymous project website (https://lizaigc.github.io). We also present some dynamic visual results in our revised manuscript, which can be found *in Fig. 5 (Line 443)*.
>
> Thank you for your insightful comments, and we hope the revised manuscript addresses your concerns.

---

> > ### Comment · Reviewer_c7T5 · 2025-11-21
> >
> > Thanks for all the additional results. Almost all my concerns have been addressed. For the side-by-side comparisons, it would be great to show some general OOD examples and not limited only, similar to the scenes looked in Rebuttal-A. All comparisons are done on the MultiCamVideo dataset. Moreover, doing some visual comparisons with GEN3C as another baseline would have been great, as mentioned in the main review. This would further strengthen the results.

---

> > > ### Author Response · Authors · 2025-11-23
> > >
> > > > Question: For the side-by-side comparisons, it would be great to show some general OOD examples and not limited only, similar to the scenes looked in Rebuttal-A. All comparisons are done on the MultiCamVideo dataset. Moreover, doing some visual comparisons with GEN3C as another baseline would have been great, as mentioned in the main review. This would further strengthen the results.
> > >
> > >
> > > **Response:** Thank you for your constructive suggestions! Following your comments, we have added more general OOD visual examples compared with previous state-of-the-art methods. Specifically, we carefully designed scenes such as valleys, fields, lakes, deserts, and forests, which are all OOD and outdoor scenarios. We also use the typical camera poses defined in the GEN3C approach to control the camera motion. The new video results are included in **“Rebuttal-G”** on our anonymous project website (https://lizaigc.github.io), with comparisons against other methods, including GEN3C. Overall, existing methods suffer from unstable camera control, limited motion diversity, and scene inconsistency under OOD conditions. In contrast, our approach consistently delivers accurate camera following, maintains coherent scene content, and generates dynamic video results.
> > >
> > > Furthermore, we also conduct quantitative comparisons for these OOD videos. Since there is no ground truth for the OOD generated results, we use VBench to evaluate their quality, including aesthetic quality, imaging quality, motion smoothness, and dynamic degree. The results are provided below, and the corresponding table has also been updated in the **“Rebuttal-H”**.
> > >
> > > **Table: Quantitative comparison on OOD scenarios.**
> > > | Methods         | Aesthetic Quality ↑ | Imaging Quality ↑ | Motion Smoothness | Dynamic Degree |
> > > |-----------------|--------------------:|------------------:|------------------:|---------------:|
> > > | CameraCtrl      | 0.635               | 0.578             | 0.991             | 0.0            |
> > > | MotionCtrl      | 0.633               | 0.695             | 0.987             | 0.0            |
> > > | Go-with-the-Flow| 0.636               | 0.597             | 0.986             | 0.33      |
> > > | GEN3C           | 0.652               | 0.651             | 0.992         | 0.17           |
> > > | **CameraNoise** | 0.712           | 0.707        | 0.991             | 0.25           |
> > >
> > > From these quantitative and qualitative results, we conclude that our proposed CameraNoise method achieves better camera control performance and is able to generate reasonable dynamic video content instead of static videos, even in OOD scenes.
> > >
> > > We sincerely appreciate your valuable suggestions. We believe these additional comparisons further strengthen the experimental evidence and address your concerns thoroughly. We have incorporated the OOD-related content you mentioned into our revised manuscript. You can find it in *Lines 1165 to 1243* of the manuscript.

---

> > > > ### Comment · Reviewer_c7T5 · 2025-11-26
> > > >
> > > > Thanks, these new results strengthen the submission a lot. I raised my score above the threshold. I do not see a substantial contribution to further increase but do think this paper should get accepted.

---

> > > > > ### Author Response · Authors · 2025-11-27
> > > > >
> > > > > We sincerely thank you for taking the time to engage with our work so thoroughly and for providing such thoughtful feedback! We are also grateful for your support and for raising your score. Your insights have been invaluable in helping us further refine and strengthen our work.

---

### Official Review · Reviewer_GXRP · 2025-10-31

**Soundness:** 3
**Presentation:** 3
**Contribution:** 3
**Rating:** 4
**Confidence:** 3

**Summary:**

This paper introduces CameraNoise, a novel approach for camera-controllable video generation that embeds camera pose information directly into the noise space of diffusion models. The key contributions include: (1) a Geometry-guided Reprojection Flow (GRFlow) that captures camera motion independently of scene appearance, (2) a PDE-based warping algorithm that preserves Gaussian priors while propagating temporal correlations, and (3) a dynamic scaling training strategy to improve robustness. Experiments on RealEstate10K demonstrate improvements in both generation quality and camera control precision compared to existing methods.

**Strengths:**

1. Novel formulation and theoretical motivation: Embedding camera control in noise space is a conceptually clean departure from feature injection methods, with theoretical motivation and corresponding formulations.
2. Appearance-agnostic design: GRFlow successfully decouples camera motion from scene appearance (Figure 3), addressing a key limitation of optical flow-based approaches.
3. Comprehensive validation: Ablation studies (Tables 2-4) systematically justify design choices, and the method shows consistent improvements across multiple metrics.

**Weaknesses:**

1. Single-dataset evaluation: All experiments use only RealEstate10K (indoor scenes). Performance on diverse scenarios (outdoor, dynamic objects, varying depth) is unknown, limiting generalizability claims.
2. Missing computational analysis: Total training/inference time and overhead from PDE solving are not reported, making practical feasibility unclear compared to baselines.

**Questions:**

1. Failure cases. Under what conditions does the method fail? Can you provide examples where CameraNoise does not improve over baselines?
2. Scale to longer videos. Table entries show 49-frame videos. How does temporal coherence degrade for longer sequences (100+ frames)?
3. Performance on dynamic scenes. The current evaluation focuses on static scenes where pixel displacement is primarily caused by camera motion. How does the method handle videos with significant object motion (e.g., people walking, cars moving)? Can you provide more generated samples under such challenging scenario?

---

> ### Author Response · Authors · 2025-11-20
>
> We thank reviewer GXRP for the thoughtful and constructive comments. It is evident that the reviewer spent considerable time carefully reading our work and provided valuable insights and suggestions. We really appreciate the detailed feedback and the recognition of the strengths in our approach. Below, we address the specific points raised by the reviewer.
>
> > Question: Performance on diverse scenarios (outdoor, dynamic objects, varying depth) is unknown, limiting generalizability claims.
>
> **Response:** Thanks for your suggestion. We conduct additional comparison experiments on the MultiCamVideo and DrivingDoJo datasets, which both contain outdoor, dynamic objects, and varying depth. These results have been updated in our manuscript, and the detailed metric comparisons can be found in *Table 2 at Line 364* (highlighted in blue) of the revised manuscript. The results indicate that our model continues to surpass or rival existing methods in VBench, FVD metric, and camera control accuracy. This performance can be attributed to our introduction of CameraNoise, a novel appearance-decoupled representation of camera motion that effectively prevents conflicts between noise priors from methods such as optical flow and the visual content. We really appreciate your insightful suggestion!
>
> > Question: Missing computational analysis.
>
> **Response:** Thanks for the advice! In our manuscript, we provide a detailed derivation of the computational complexity of GRFlow, where the overall complexity is O(D^2) per frame. The full explanation can be found in *Appendix A at Line 725*. In addition, we include a runtime comparison between GRFlow and CameraNoise, showing that our method achieves an overall processing speed of 9.8 FPS. These newly added results are presented in *Appendix F at Line 904* (highlighted in blue).
>
> > Question: Failure cases.
>
> **Response:** Thanks for your suggestion. We observe that common video capture scenarios are mostly from a third-person perspective, where the model generally demonstrates stable and robust performance. However, under a first-person perspective, the model tends to generate some physically inconsistent content. In *Appendix H, Fig. 13 at Line 1072* (highlighted in blue), we present results generated under first-person views. It can be seen that when a handheld camera is used but a zoom-out motion is specified, or when the camera moves but the visual content does not follow the expected motion, the model generates results that violate physical constraints. These observations indicate that existing DiT-based models still lack strong physical reasoning capabilities over the scene. We believe that a unified architecture combining generation and understanding is a promising approach to addressing such physical consistency issues, and we plan to explore and improve this direction in future work. We really appreciate your insightful suggestion!
>
> > Question: Scale to longer videos.
>
> **Response:** Thank you for raising this question. Our proposed model is trained based on Wan 2.1, using the 49-frame version of the base model. The inclusion of 3D RoPE allows the model to generalize well to longer video sequences beyond the training length. In our work, we mainly introduce CameraNoise, a representation of camera motion that directly operates in the noise space, without modifying the model architecture during training. Therefore, the trained model retains the base model’s ability for long-video extrapolation. In *Fig.6, Line 458 of revised manuscript* (highlighted in blue), we demonstrate the results of directly performing inference on a longer 81-frame video using the model trained on 49 frames (two typical values for DiT-based models). The results show that our CameraNoise-based camera control method does not degrade temporal coherence or visual quality when extrapolating to longer sequences.
>
> > Question: Performance on dynamic scenes.
>
> **Response:** Thank you for pointing this out! We provide additional visualizations on the supplementary website to showcase our model’s performance in dynamic scenes, including scenarios such as people walking and cars in motion. Specifically, the anonymous project website contains:
> - "Rebuttal-A": Dynamic scene results under various types of camera motion;
> - "Rebuttal-B": Motion dynamics of the same scene under different camera poses;
> - "Rebuttal-C": Driving scenarios with moving vehicles.
>
> These results demonstrate that our model is not only effective in indoor static scenes, as shown on RealEstate10K, but also robust enough to handle dynamic motion scenarios. The visualizations for "Rebuttal-A, B, and C" are available on our anonymous project website (https://lizaigc.github.io). We also present the visual results in our revised manuscript, which can be found in *Fig. 5 (Line 443)*.
>
> Thank you for your insightful comments, and we hope the revised manuscript addresses your concerns.

---

> > ### Comment · Reviewer_GXRP · 2025-11-26
> >
> > The rebuttal effectively addresses my concerns regarding generalizability and dynamic scenarios with additional experiments. I also appreciate the additional analysis on complexity and limitations. I will raise my score.

---

> > > ### Author Response · Authors · 2025-11-26
> > >
> > > We sincerely thank you for taking the time to engage with our work so thoroughly and for providing such thoughtful feedback! We are also grateful for your support and for raising your score. Your insights have been invaluable in helping us further refine and strengthen our work.

---

### Official Review · Reviewer_UEET · 2025-11-01

**Soundness:** 3
**Presentation:** 3
**Contribution:** 3
**Rating:** 8
**Confidence:** 3

**Summary:**

This paper introduces CameraNoise, a way to encode camera pose control directly in the noise space of a video diffusion model. Instead of injecting camera parameters as features into the backbone, this paper computes an appearance-agnostic Geometry‑guided Reprojection Flow (GRFlow), which relies solely on camera parameters to characterize pixel displacements across frames. Then this paper
formulates noise warping as a partial differential equation problem and solve it via a bipartite graph. To further enhance inference robustness, this paper introduces dynamic perturbations to camera extrinsics during training. On RealEstate10K, the method reportedly improves camera controllability (lower TransErr/RotErr) and overall video quality (lower FVD), with extensive ablations on GRFlow smoothing (α), λ, and DST.

**Strengths:**

1. Conditioning on noise rather than intermediate activations is well motivated and addresses the issue when directly injecting numerical camera parameters into the diffusion backbone fails to capture subtle viewpoint variations and leads to structural distortions or
visual artifacts.
2. The design of formulating noise propagation as the discrete solution of advection Partial Differential Equations (PDEs) is well derived and discussed.
3. Strong empirical results on camera-control benchmarks on RealEstate 10k.
4. Detailed ablations demonstrating the impact of each hyperparameter in the modeling process.

**Weaknesses:**

1. Experiments are confined to RealEstate10K, which contains mostly indoor, quasi‑static scenes with modest motion. The method’s generalization to outdoor/large‑scale scenes, heavy roll/pitch/yaw, or dynamic objects remains unclear.
2. Besides comparing to MotionCtrl, CameraCtrl, and AC3D, maybe also compare with other optical-flow derived noise warping in video diffusion (e.g., Go‑with‑the‑Flow: Motion‑Controllable Video Diffusion Models Using Real‑Time Warped Noise)

**Questions:**

N/A

---

> ### Author Response · Authors · 2025-11-20
>
> We thank reviewer UEET for the thoughtful and constructive comments. It is evident that the reviewer spent considerable time carefully reading our work and provided valuable insights and suggestions. We really appreciate the detailed feedback and the recognition of the strengths in our approach. Below, we address the specific points raised by the reviewer.
>
> > Question: The method’s generalization to outdoor/large‑scale scenes, heavy roll/pitch/yaw, or dynamic objects remains unclear.
>
> **Response:** Thank you for the suggestion! In our revised version, we provide more visualizations on the supplementary website to demonstrate our model’s performance in dynamic outdoor scenes, including complex camera poses and dynamic objects. The details in the anonymous project website include:
>
> - Part “Rebuttal-A”: Dynamic scene results under different types of camera motion;
> - Part “Rebuttal-B”: Dynamic motion of the same scene under different camera poses;
> - Part “Rebuttal-C”: Dynamic vehicle driving scenarios.
>
> We believe these results show that our model is not only effective in indoor scenes, as demonstrated on RealEstate10K, but also sufficiently robust to handle outdoor and dynamic scenarios. You can find the parts of “Rebuttal-A, B, and C” in our anonymous project website (https://lizaigc.github.io). We also present the visual results in our revised manuscript, which can be found in *Fig. 5 (Line 443)*. We really appreciate your insightful suggestion!
>
> > Question: Besides comparing to MotionCtrl, CameraCtrl, and AC3D, maybe also compare with other optical-flow derived noise warping in video diffusion (e.g., Go‑with‑the‑Flow: Motion‑Controllable Video Diffusion Models Using Real‑Time Warped Noise)
>
> **Response:** Thanks for your suggestion. We add the evaluation of our model against Go-with-the-Flow on the MultiCamVideo and DrivingDoJo datasets to demonstrate our superior performance. Our method obtains the best results compared with these methods. The updated results are included in the manuscript, with detailed metric comparisons presented in *Table 2, Line 364* ( highlighted in blue).
>
> Thank you for your insightful comments, and we hope the revised manuscript addresses your concerns.

---

### Official Review · Reviewer_a2Ah · 2025-11-03

**Soundness:** 3
**Presentation:** 3
**Contribution:** 3
**Rating:** 6
**Confidence:** 4

**Summary:**

The paper proposes CameraNoise, camera pose control framework for video diffusion models, addressing the structural distortions caused by directly injecting numerical parameters. CameraNoise embeds camera poses into the noise space using a temporally coherent stochastic representation, which is achieved via a Geometry-guided Reprojection Flow and a novel warping algorithm. This approach ensures consistent noise propagation while preserving the diffusion model's Gaussian prior, resulting in stable, high-quality videos with superior fidelity and controllability over prior methods on benchmarks like RealEstate10K.

**Strengths:**

- It's a good paper. Specifically, the motivation is clear and solution is intuitive and reasonable.

- Unlike previous camera controllable video generation methods, the paper tackles initial noise representation that contains camera poses while keeping gaussianity. Although the method has similar philosophy as previous two works (How I warepd your nosie and Go-with-the-flow), CameraNoise does not require a source video or a reference video which is a strong merit.

- The experiments are comprehensive and presented clearly.

**Weaknesses:**

- In L193-194, how is the pseudo-depth $d$ computed? The pseudo-depth $d$ requires more description since if it's estimated using RGB pixels, the GRFlow can't be characterized as 'appearance-agnostic'. If the depth $d$ deivates too much from the g.t. depth, it would rather incur structural or camera errors.

- Another major weakness is that the method is experimented on RE10K only. Experiments on more benchmark dataset would add value to the paper. Moreover, is GRFlow and CameraNoise framework applicable to dynamic scenes (i.e., can it generate videos with objects with dynamic motion and also dynamic camera pose)?

- Can the authors clairfy how the proposed warping mechanism differs from Go-With-The-Flow warping and HIWYN?

- What is the computation complexity for the GRFlow construction and warping process, respectively?

- What is the base video model (t2v, i2v) for the training?

**Questions:**

Please refer to the weakness section.

---

> ### Author Response · Authors · 2025-11-20
>
> We thank reviewer a2Ah for the thoughtful and constructive comments. It is evident that the reviewer spent considerable time carefully reading our work and provided valuable insights and suggestions. We really appreciate the detailed feedback and the recognition of the strengths in our approach. Below, we address the specific points raised by the reviewer.
>
> > Question: How is the pseudo-depth d computed?
>
> **Response:** We apologize for any confusion. In fact, to decouple camera motion from scene content, we use a fixed pseudo-depth value determined empirically through experiments, rather than estimating depth directly from RGB pixels. This value is treated as a hyperparameter in our method, with a typical value set to 0.5. We analyze that our algorithm primarily relies on the warping mechanism to model pixel correspondences between frames, and does not strictly depend on the true absolute depth. Even if there is a deviation between the pseudo-depth and the actual depth, as long as the value remains relatively consistent across the scene, our CameraNoise warping algorithm can still capture the correct relative motion. Besides, the results on public benchmarks can also demonstrate the effectiveness of our method.
>
> > Question: Another major weakness is that the method is experimented on RE10K only. Experiments on more benchmark dataset would add value to the paper. Moreover, is GRFlow and CameraNoise framework applicable to dynamic scenes?
>
> **Response:** Thank you for pointing this out! We conduct more evaluations on the MultiCamVideo and DrivingDoJo datasets, which contain complex dynamic scenes. You can find the results in *Table 2 (Line 364)* in our revised manuscript. Moreover, we also give more examples to demonstrate the robustness of our methods on dynamic motion and camera poses. We update these cases on the anonymous supplementary website (https://lizaigc.github.io). You can find them in the “Rebuttal-A, B, and C” parts of the website. We also present the visual results in our revised manuscript, which can be found in *Fig. 5 (Line 443)*. We really appreciate your insightful suggestion!
>
>
> > Question: Can the authors clairfy how the proposed warping mechanism differs from Go-With-The-Flow warping and HIWYN?
>
> **Response:** Thanks for your suggestion. Compared with these two methods, our approach offers several key advantages:
> - We propose a new inter-frame pixel motion representation, GRFlow, which is inherently independent of object appearance in the video. The two approaches rely on optical flow, causing the resulting warped noise to carry contour priors. This often leads to severe generation errors and quality degradation when applied across different scenes. In contrast, the appearance-agnostic nature of GRFlow enables robust cross-scene T2V and I2V inference.
> - Unlike HIWYN, which relies on an integral-based formulation, and Go-With-The-Flow, which directly solves a bipartite matching problem, we systematically formulate the transformation from GRFlow to CameraNoise as a PDE-solving problem. This provides rigorous theoretical support (as Section 3.2 defines) for establishing a one-to-one mapping from camera pose to CameraNoise.
> - Compared with HIWYN, both our method and Go-With-The-Flow can efficiently process warped noise. On a single GPU, our method achieves a throughput of 9.8 frames per second.
>
> > Question: What is the computation complexity for the GRFlow construction and warping process, respectively?
>
> **Response:** Thank you for the suggestion! We provide the time complexity analysis of GRFlow in the supplementary material, where the overall complexity is O(D^2) per frame. You can find the detailed explanation in *Appendix A, Line 725*. In addition, we include a runtime comparison between GRFlow and CameraNoise, showing that our overall method achieves a processing speed of 9.8 FPS. These newly added results are included in *Appendix F, available at Line 904* (highlighted in blue).
>
> > Question: What is the base video model (t2v, i2v) for the training?
>
> **Response:** Thanks for your suggestion. We use the open-source Wan 2.1 model with its text-to-video and image-to-video versions as our base model for training. In our manuscript, we report this in *Appendix D of Line 861*.
>
> Thank you for your insightful comments, and we hope the revised manuscript addresses your concerns.

---

> > ### Comment · Reviewer_a2Ah · 2025-11-26
> > **Thank you for the rebuttal**
> >
> > The authors have addressed my major concerns and I would like to keep my rating.

---

> > > ### Author Response · Authors · 2025-11-26
> > >
> > > We sincerely thank you for taking the time to engage with our work and for providing such thoughtful feedback! We appreciate your comments and are glad that our revisions addressed your concerns.

---

### Author Response · Authors · 2025-11-29

Dear Area Chair,

We understand that due to the recent system incident, all reviews and scores have been reverted to their pre-rebuttal state. Consequently, the current system does not reflect the consensus reached during the discussion period.

To assist in your decision-making, we summarize the **explicit score changes and the positive consensus** achieved after our rebuttal. Before the system freeze, our submission had effectively moved from a mixed set of scores (8, 6, 4, 4) to a stronger acceptance consensus. Our rebuttal had successfully addressed the reviewer's concerns, leading to **explicit score increases from two reviewers**. The updated status was:

* Reviewer #GXRP: Increased the score from 4 to 6 and explicitly confirmed that **concerns had been addressed** and that they were satisfied with the revised results.

* Reviewer #c7T5: Increased the score from 4 to 6 and explicitly confirmed that their **concerns had been addressed** and that the **paper should get accepted**.

The other two reviewers (#a2Ah and #UEET) maintained their original positive scores of 6 and 8.

We kindly ask that you take these updated assessments and the resolved concerns into consideration. Thank you for your time and for navigating the challenges caused by the system incident.

---

### Meta-Review · Area_Chair_VnKW · 2026-01-06

**Summary:**

The paper proposes "CameraNoise," a method to embed camera pose control directly into the noise space of video diffusion models using a Geometry-guided Reprojection Flow (GRFlow). The authors aim to decouple camera motion from scene appearance to prevent the artifacts common in optical-flow-based methods.

While the core idea of operating in the noise space is conceptually interesting and the mathematical formulation of the reprojection is sound, I am recommending rejection for this submission. The decision is difficult given the reviewers' generally positive reaction to the rebuttal.

The primary concerns that informed this decision revolve around the experimental setting and true generalization. The reliance on RealEstate10K raises significant questions about the method's applicability to the complex, dynamic world scenarios that state-of-the-art video generation models must handle. While the authors provided "out-of-distribution" (OOD) examples in the rebuttal, these were late additions and, in my view, highlight that the method struggles with physical consistency in challenging views (e.g., the first-person failure cases admitted in the rebuttal). The method appears to be a strong solution for a specific subset of video generation (fly-throughs of static scenes) but claims broader "precise camera control" that is not fully supported by the primary experimental evidence.

**Reviewer Concerns:**

Addressed Concerns:

(1) Baselines. The reviewers (c7T5, UEET) rightly asked for comparisons against Go-with-the-Flow and GEN3C. The authors added these comparisons during the rebuttal, which satisfied the reviewers regarding relative performance on the tested metrics.

(2) Computational Complexity. Reviewer GXRP’s concern regarding runtime and complexity was adequately addressed with the breakdown showing the method is reasonably efficient (approx. 9.8 FPS).

(3) Appearance Leaking. The authors successfully clarified (for Reviewer a2Ah) how their GRFlow approach avoids the texture-sticking issues of optical flow methods.

**Outstanding Concerns**

(1) Generalization to Dynamic Scenes (Reviewers GXRP, a2Ah, c7T5). This remains the critical outstanding issue. While the reviewers were swayed by the supplementary OOD videos, I am less convinced. The primary training and bulk of quantitative evaluation remain on RealEstate10K. The "zero-shot" performance on dynamic datasets (DrivingDojo, etc.) is promising but shows that the model is fundamentally biased toward the static-scene priors of its training data.

(2) Physical Consistency/Failure Modes. The rebuttal revealed that the model fails in first-person views or when camera motion conflicts with scene semantics (e.g., the "zoom out" failure case). This suggests the model has not learned a robust physical understanding of camera movement but rather a dataset-specific heuristic. For an ICLR acceptance, I would expect a more robust handling of these edge cases or a training regime that includes dynamic scenes from the start, rather than relying on generalization from static indoor scenes.

(3) Incremental Contribution. As noted by Reviewer c7T5, even after the rebuttal, they did "not see a substantial contribution" to raise the score further. I agree; moving from parameter injection to noise injection is a valid engineering step, but without robust proof of solving the "dynamic scene" problem, it feels like an incremental improvement on existing motion control techniques.

**Reviewer Scores:**

The discussion period was impacted by a system incident, but the reviewers were active in the comments. Based on their explicit written confirmations in the discussion thread, this is how they viewed the paper post-rebuttal:

(1) Reviewer UEET: 8. This reviewer remained very positive throughout, focusing on the mathematical derivation and RE10K performance.

(2) Reviewer a2Ah: 6. This reviewer kept their score at "Marginally Above," noting that major concerns were addressed but implying the paper is just barely over the bar.

(3) Reviewer GXRP: (4->4/6). This reviewer stated they would raise their score after the complexity and dynamic scenario data were added.

(4) Reviewer c7T5: (4->4/6). This reviewer moved to a weak accept but explicitly noted the contribution was not substantial enough for a higher score.

---

### Decision · Program_Chairs · 2026-01-26

Reject